# Identifying attacks in the Russia–Ukraine conflict using seismic array data

Ben D. E. Dando[1✉], Bettina P. Goertz-Allmann[1], Quentin Brissaud[1], Andreas Köhler[1,2], Johannes Schweitzer[1,3], Tormod Kværna[1] & Alexander Liashchuk[4]

Seismometers are generally used by the research community to study local or distant earthquakes, but seismograms also contain critical observations from regional[1,2] and global explosions[3], which can be used to better understand conflicts and identify potential breaches of international law. Although seismic, infrasound and hydroacoustic technology is used by the International Monitoring System[4] to monitor nuclear explosions as part of the Comprehensive Nuclear-Test-Ban Treaty, the detection and location of lower-yield military attacks requires a network of sensors much closer to the source of the explosions. Obtaining comprehensive and objective data that can be used to effectively monitor an active conflict zone therefore remains a substantial challenge. Here we show how seismic waves generated by explosions in northern Ukraine and recorded by a local network of seismometers can be used to automatically identify individual attacks in close to real time, providing an unprecedented view of an active conflict zone. Between February and November 2022, we observed more than 1,200 explosions from the Kyiv, Zhytomyr and Chernihiv provinces, providing accurate origin times, locations and magnitudes. We identify a range of seismoacoustic signals associated with various types of military attack, with the resulting catalogue of explosions far exceeding the number of publicly reported attacks. Our results demonstrate that seismic data can be an effective tool for objective monitoring of a continuing conflict, providing invaluable information about potential breaches of international law.

Although media reports show the devastation associated with the war in Ukraine, obtaining a comprehensive and unbiased overview of the continuing military attacks is a substantial challenge. Social media posts and traditional media outlets all have the potential to be subjective and are, in fact, often subject to manipulation for the purpose of misinformation and propaganda. Having a more complete and objective picture showing exactly where and when attacks are taking place is vital for developing a clear understanding of the scale of a conflict, how it is progressing and identifying potential breaches of international law.

## Methods of conflict monitoring

Satellite imagery has been shown to be an effective means of providing high-resolution pictures of military attacks in Ukraine. Such data are now accessible to members of the public[5] and help to support the emerging open-source intelligence community. However, although open satellite images can provide high spatial resolution, previous knowledge is required for the time and location of the imagery. Providing comprehensive coverage across a large region in real time is beyond the capability of this technology and it thus suffers from the incompleteness that plagues traditional reporting.

Yet, satellite images are not the only source of objective conflict data. The seismic and sound waves that are generated by an explosion can propagate over hundreds of kilometres, at velocities of up to approximately 8 km s[−1] in the ground and approximately 0.34 km s[−1] in the air. These signals can be recorded by seismometers and microbarometers at high sampling rates (typically between 40 and 200 Hz), which can help monitor a conflict in real time. Acoustic and seismic ranging methods have been used from as early as the First World War for locating artillery positions and were fundamental to the development of modern-day seismic exploration methods[6–8]. Methods have since evolved to focus either on locating artillery positions and impact areas using acoustic sensors from experimental datasets[9] or to infer properties of individual large explosions[1,2,10]. However, the real-time analysis of seismic and acoustic signals from an active military conflict is, so far, absent in the literature, in part owing to a lack of suitable data.

The availability of seismic and infrasound data, however, is now at its most prevalent. The International Monitoring System (IMS), a global network to detect nuclear explosions as part of the Comprehensive Nuclear-Test-Ban Treaty, now stands at more than 200 seismic and infrasound stations combined[4]. Similarly, openly available data used for monitoring and researching earthquakes and the Earth's interior is vast and continues to expand[11]. These sensors can be used for conflict monitoring, with the performance of such a network being based on the proximity of the sensors to the region being monitored, the size of the explosions and how well the energy is transmitted.

[1]NORSAR, Kjeller, Norway. [2]The Arctic University of Norway – UiT, Tromsø, Norway. [3]University of Oslo, Oslo, Norway. [4]Main Centre of Special Monitoring, National Space Facilities Control and Testing Centre, State Space Agency of Ukraine, Gorodok, Ukraine. ✉e-mail: ben@norsar.no

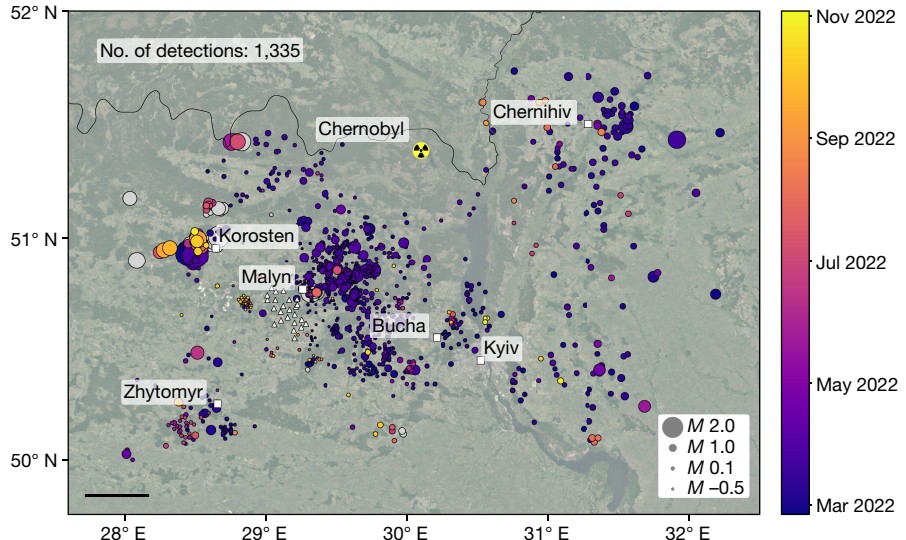

**Fig. 1 | Map of automatic seismic detections (circles) between January and November 2022.** Detections are coloured by date of occurrence and scaled by magnitude. Detections before the start of the invasion on 24 February are shown in grey. Locations of individual seismic sensors of the Malyn array are shown by white triangles. The location of the Chernobyl nuclear power plant is indicated. Scale bar, 50 km.

## Application to Ukraine

Approximately 100 km northwest of Kyiv, Ukraine, the IMS has a seismic array operated by the Ukrainian National Data Centre, denoted Malyn AKASG (treaty code: PS45). It comprises 23 vertical-component broadband seismometers and a single three-component broadband seismometer, with an aperture of approximately 27 km, with around 2 km between each sensor.

The intended design was for the detection of nuclear tests at teleseismic distances (>3,300 km) with classical array processing techniques. Under the assumption that the incoming wavefronts are planar and originate from distances much larger than the array aperture, coherent signals across the array are stacked to enhance the signal-to-noise ratio, whereas the time delays between the individual array elements are used to estimate the direction of an incoming wavefront[12]. To direct the array towards detecting local and regional seismic activity, we must abandon the classical plane-wave assumption and use observations from individual seismometers to accurately locate events near the array. With the large spatial footprint and high sensor count, the Malyn array offers a unique opportunity for monitoring conflict-related explosions throughout the Zhytomyr, Kyiv and Chernihiv provinces in northern Ukraine.

We have implemented a continuous monitoring solution that can automatically detect and locate explosions, using seismic signals recorded at the Malyn array. Data are continuously transmitted to the International Data Centre (IDC) in Vienna and from there to Norway for automatic processing, generating results close to real time. Our implementation builds on a methodology designed for automatically detecting and locating microseismic activity[13] using a migration/stacking approach[14–16] applied to characteristic functions of the short-term average to long-term average amplitude ratio (STA/LTA), tailored for detecting P-wave and S-wave signals at each sensor (Methods). Despite observing signals from air-to-ground acoustic waves, we initially omit these signals from the automatic location algorithm as they are observed infrequently and their less impulsive signals lead to a higher number of false detections and a reduced sensitivity of the detection algorithm. For the seismic signals, however, we can produce high-precision event locations and timings for the region northwest of Kyiv, allowing us to observe the development of the conflict with unprecedented accuracy.

Although muzzle blasts, ballistic shock waves and the impact detonation can all generate infrasound energy[9], it is only the impact detonation that is most likely to generate sufficient seismic energy that can be observed at the distances we monitor[17,18]. The infrasound waves that we observe at the seismometers resulting from the air-to-ground coupling travel at much lower velocities than the seismic waves, which ensures that they do not adversely affect our seismic detection algorithm. We are therefore confident that our detections most probably correspond to impact explosions.

## Detected explosions in northern Ukraine

We have automatically detected and located 1,282 explosions from 24 February to 3 November 2022 in a region of approximately 300 km × 222 km around the Malyn array, including parts of the Zhytomyr, Kyiv and Chernihiv provinces (Fig. 1). A video of the detected explosions is provided in the Supplementary Information. Owing to the detection bias close to the array, the magnitude of completeness reduces with distance from the array, meaning that the lowest-magnitude explosions cannot be detected in locations such as Chernihiv, which is approximately 170 km from the array. To establish a detection baseline, we have also processed data before the beginning of the war, from 1 January 2022, which increases our catalogue by a further 53 daytime explosions, associated with mining and quarry activity in the region (Fig. 2). Post-invasion, we observe clusters of activity around the metropolitan areas of Zhytomyr, Korosten, northwest of Kyiv, Chernihiv and Malyn. The most prominent activity is to the northeast of Malyn, which—although corresponding to a region in which detection capability is high—also coincides with a region of intense fighting at the limits of the Russian-controlled territory during late February and March. Between 24 February and 31 March, we observe an average of 29 detected explosions per day, with the highest activity on 7 March, for which 64 explosions are identified. Russian-controlled territory in the region expanded until 21 March, after which there were claimed Ukrainian counter-offensives before the final Russian withdrawal from the Kyiv region was reported on 2 April (ref. 19). We observe the last heavy bombardment on 31 March, with only two explosions detected on 1 April. After the reported Russian withdrawal, the background activity largely returns to pre-invasion levels, with the resumption of some mining activity. However, sporadic targeted attacks at strategic

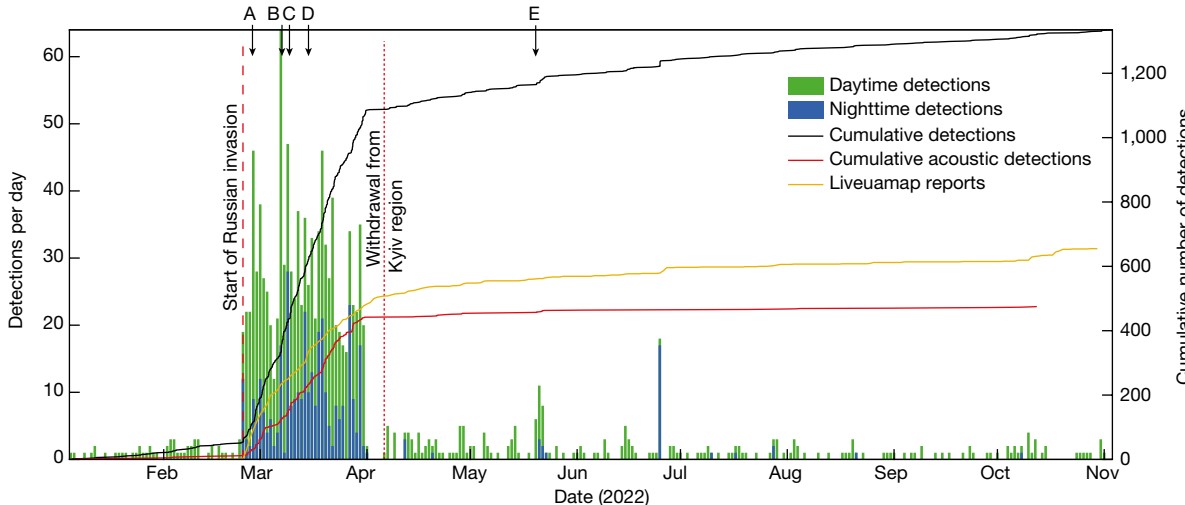

**Fig. 2 | Timeline of automatic seismic detections.** The histogram distribution separates daily detections during nighttime (blue) and daytime (green). The black line shows the cumulative number of detections, whereas the red line shows the cumulative events with observable acoustic signals. The cumulative number of reported Liveuamap events is shown by the orange line. The start of the Russian invasion is indicated by the dashed red line and the reported Russian withdrawal from the Kyiv region is shown by the dotted red line. Labelled arrows point to specific example attacks shown in Fig. 4.

locations continue. The progression from pre-invasion mining explosions to intense military attacks, corresponding to the advancing Russian military, followed by the infrequent but targeted attacks, can be observed in Fig. 3.

Most explosions associated with a military conflict occur at or above ground level, releasing most of their energy into the atmosphere. The resulting acoustic waves provide key constraints on the explosion source parameters[20]. Although we choose to omit the acoustic waves from our automatic location algorithm, we can still attempt to identify these arrivals based on the seismic detections. By stacking the seismic envelope in a time window constrained by waves travelling at acoustic velocities (Methods), we identified clear acoustic signatures for 29% of the seismic events (red line in Fig. 2 and green dots in Fig. 4f), which are able to further constrain both spatial and temporal event information (inset maps in Fig. 4a,e). The absence of acoustic waves for most of the events may be explained in part by the choice of acoustic detection threshold, the absorption and scattering processes that are notable at high frequencies and the absence of wind waveguides close to the ground, preventing the acoustic energy from efficiently propagating along the surface[21]. It is also worth noting that not all explosions observed in the waveform data feature seismic arrivals, with some events only detectable from their acoustic signature. Such events most probably correspond to explosions at higher altitude, at larger distances from the source (>100 km) or with lower yield. These observations highlight that both acoustic and seismic monitoring can play an important role in conflict monitoring.

## Data validation

Part of the value of the detected explosions lies in being able to use them for either validation of reported explosions or to provide completely new information for unreported explosions. As an example, on Friday 20 May at 09:37 UTC, the mayor of Malyn (100 km northwest of Kyiv) released a video message stating that there had been a missile attack on the town. Subsequent photographs published in the media showed damage to the railway tracks and reports from the Russian Ministry of Defence stated that the station was deliberately targeted[22]. Almost 4 h before these reports, we had automatically identified three co-located explosions at 05:39:59, 05:40:11 and 05:40:23 UTC. Although our automatic location estimate was 1.4 km from the resulting crater, manual analysis of the signals was able to locate them to within 100 m of this site (Fig. 4e and Extended Data Table 2).

Further explosion examples with different waveform characteristics are shown in Fig. 4: the Hostomel Airport attack (Fig. 4a) also shows observable infrasound arrivals that can be used to improve the event location and examples in Fig. 4b–d show events without observable acoustic arrivals. Example spectrograms show clear differences between different phase arrivals, including the dispersive nature of the surface wave (Rg) arrival.

To compare our explosion catalogue with publicly reported attacks, we collected conflict data provided by the Live Universal Awareness Map (Liveuamap; https://liveuamap.com/) in the same region (Methods). This platform aggregates reported events from various media outlets using artificial intelligence, which are manually verified. We find a very similar trend in the timeline of the reported events compared with the explosion catalogue, with spikes in activity both after the initial invasion and for specific targeted attacks after the Russian withdrawal from the region in April 2022 (Fig. 2). With the exception of three days (25–27 February 2022), the number of detected explosions always exceeds the number of reported attacks in this region for the most active time period (February–March). Despite the general agreement between reported and seismic events, it is worth noting that large uncertainties exist in both the timing and the locations of the reported data, as they are generally based on anecdotal data sources.

As with all automatic detection algorithms, our method is not exempt from false detections. We used a fixed detection threshold that was kept relatively low to improve true detection rates, at the expense of more false positives. False positives include large signals from events originating outside our monitoring region, such as distant earthquakes or explosions, that provide sufficient coherency to generate aliased locations in our monitoring region. Moreover, when several explosions are recorded in short succession, signals from the different explosions can be incorrectly attributed to the different events, resulting in both the misidentification of the number of explosions and the generation of mislocations. Such false positives can, in part, be mitigated by pre-processing steps and careful parameter selection, but these are also subject to the same trade-off between detectability and false-positive detection. As a result, we performed a further manual screening of the automatic results to reduce the number of false positives.

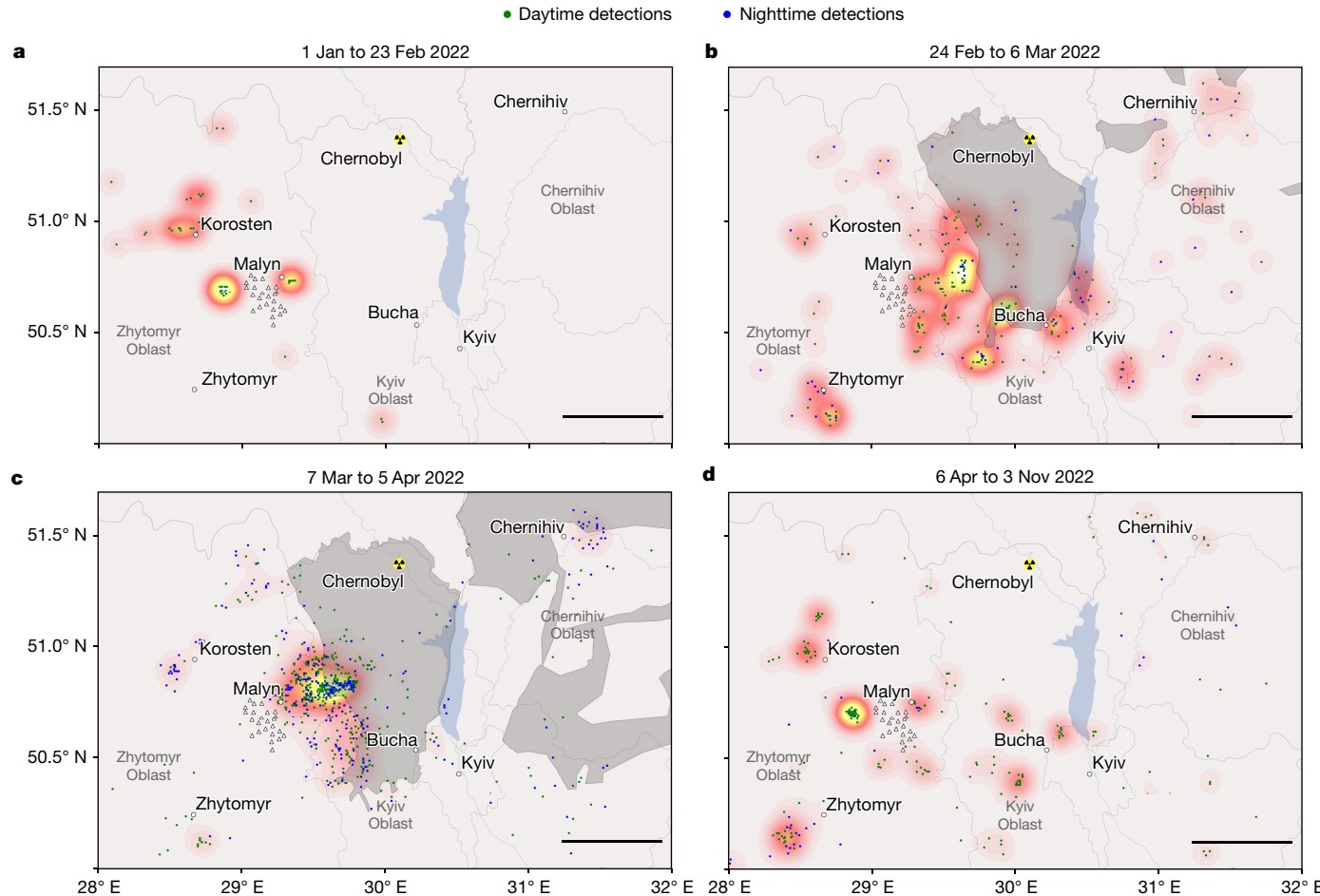

**Fig. 3 | Density plots of the automatic seismic detections for various time periods.** Individual detections are shown by blue (nighttime) and green (daytime) dots. The location of the Malyn seismic array is shown by the white triangles. Each panel is scaled to its maximum value. **a**, Pre-invasion period from 1 January to 23 February 2022, with a maximum event density of 0.09 events km⁻². **b**, Immediately after the Russian invasion from 24 February to 6 March 2022, with a maximum event density of 0.11 events km⁻². **c**, Period of intense fighting from 7 March to 5 April 2022, with a maximum event density of 0.73 events km⁻². **d**, Period after the reported Russian withdrawal (2 April 2022) from 6 April to 3 November 2022, with a maximum event density of 0.25 events km⁻². The grey shaded areas in **b** and **c** indicate regions occupied by Russian troops. Scale bars, 50 km.

## Explosive size

Estimating the explosive yield from seismic data is a challenging research area, with numerous approaches based on both empirical observations and physics-based models[23–25]. Recent methods combining both seismic and acoustic observations[26] show great promise in resolving both yield and height of explosions. However, because the Malyn array comprises vertical-component data on all but a single site, we are limited in the approach we can take. We focus on providing a rapid evaluation of the explosive strength by automatically computing seismic magnitudes (Methods). Empirical relationships between explosive yield and seismic magnitude are well established for underground nuclear explosions at specific test sites[27–29], but these can be poor analogues for surface explosions, in which there are substantial differences in coupling and energy propagation. However, together with a catalogue of land-based explosions with known yields[30], we estimate upper and lower limits of the yield for each magnitude. We observe automated local magnitudes between −1.25 and 2.24 (Fig. 1). Spot checks of the automated magnitudes using manual analysis provide consistent values within approximately 0.3 magnitude units, although the lower-magnitude estimates seem closer to −0.6 in the manual analysis, corresponding to an explosive yield of between 0.03–9.00 kg TNT. For comparison, the explosive yield of an OF45 152-mm projectile used by Russian

howitzers is 7.65 kg TNT[31], suggesting that the upper estimate is more realistic. For the largest-magnitude events ($M > 1.7$), these are associated with mining and quarry activity close to Korosten (Fig. 1). The largest explosion that can be clearly associated with a military attack has a magnitude of 1.7 and corresponds to an air strike that targeted Chernihiv on 10 March 2022 (Fig. 4c). The explosive yield for this explosion is estimated to between 352 and 3,083 kg. Considering an Iskander ballistic missile has a yield of approximately 700 kg (ref. 32), our maximum yield estimate is much too high, but the lower estimate is certainly feasible for such an air strike. To further improve yield estimates, we also derive independent yield estimates from acoustic phase amplitudes (Methods). However, acoustic-based prediction models[33] lead to even larger yields, highlighting the need for yield calibration experiments.

## A new tool for conflict monitoring

The analysis of seismic data collected during the 2022 Russia–Ukraine war has demonstrated the first known case of using seismic data to monitor a conflict in near real time. The distribution of the detected military-related explosions corresponds well to zones of intense military activity or individual artillery and missile strikes. Although our catalogue of explosions is not exhaustive, we demonstrate a comprehensiveness that exceeds the number of publicly

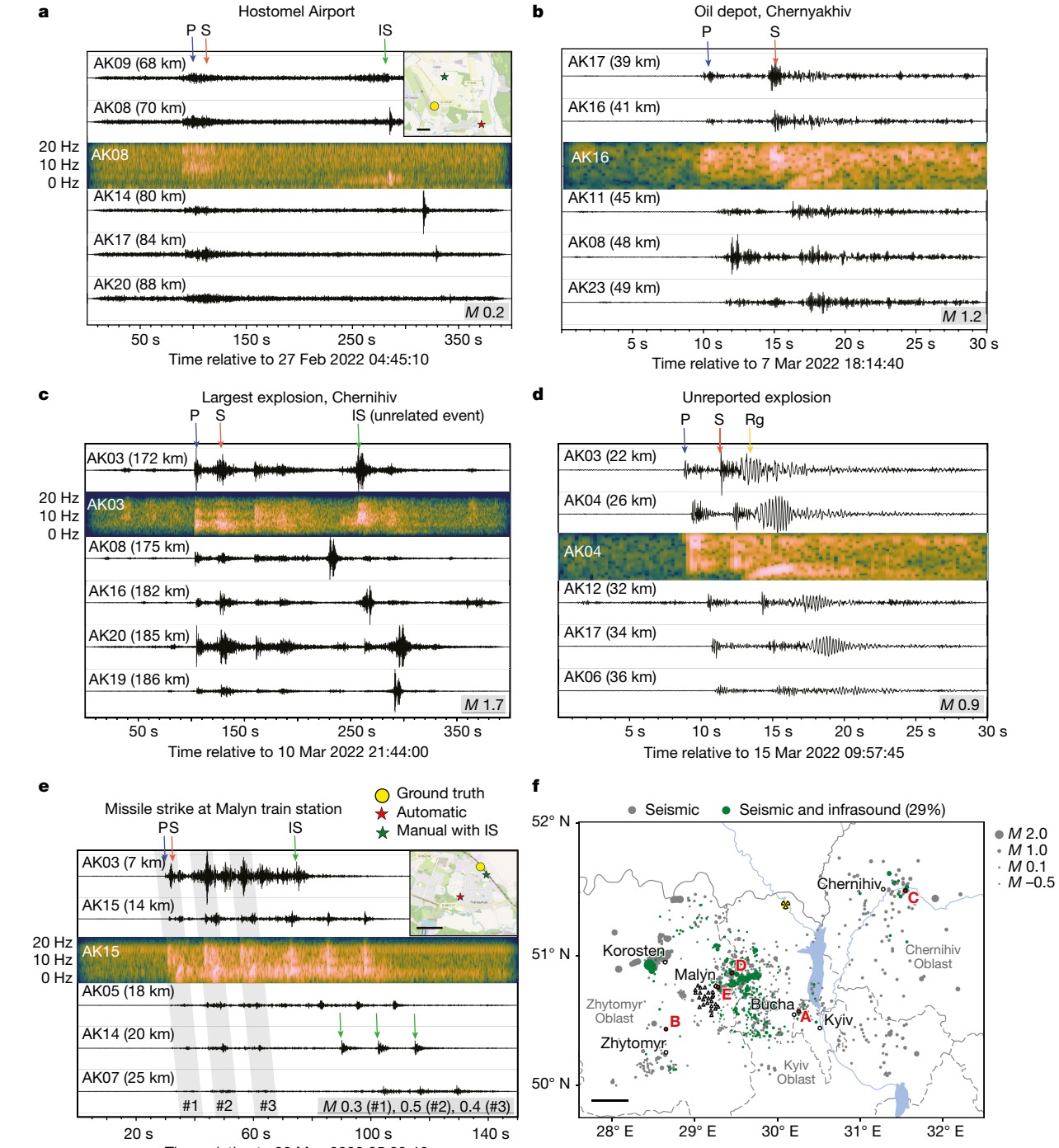

**Fig. 4 | Data examples from individual explosions. a**, Hostomel Airport attack. **b**, An explosion at an oil depot in Chernyakhiv. **c**, An airstrike on Chernihiv, the largest military explosion recorded. **d**, An unreported explosion northeast of Malyn. **e**, A missile strike at Malyn train station. Selected waveforms are shown at various distance ranges from the source. One spectrogram example per event is shown below the waveform of the respective station. P-wave (P), S-wave (S), acoustic wave (IS) and surface wave (Rg) are labelled by arrows in blue, red, green and yellow, respectively. Event magnitudes are indicated. Inset maps in **a** and **e** (scale bars, 1 km) show the automatic event location (red star) and the manual event location including acoustic arrivals (green star) compared with the ground-truth location (yellow circle). These locations are provided in Extended Data Table 2. **f**, Map indicating the example event locations as red circles and labelled A–E. Green circles show all events with detected acoustic arrivals and grey circles without observable acoustic arrivals. Scale bar, 50 km.

reported attacks, demonstrating its value in both report verification and as an original data source. Our automatic seismic-phase detection method provides accurate spatial (<5 km error) and timing (<1 s error) information about regional events in northern Ukraine (<100 km from the Malyn array). The automatic detection and analysis of acoustic phases in post-processing allows to further improve spatial accuracy. The same methodology can be applied to other arrays or dense sensor networks in the vicinity of conflicts. This unique dataset also provides the opportunity for the automatic characterization of artillery or ammunition types, allowing improved scrutiny of the conflict and helping to determine breaches of international law.

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

## Methods

### Malyn AKASG (PS45)

The detection of explosions in this study (Supplementary Table 1) was performed using data collected at the Ukrainian primary seismic station of the IMS, which operates as part of the Comprehensive Nuclear-Test-Ban Treaty Organization (CTBTO). The station is denoted as Malyn AKASG with the treaty code PS45. Details of all 24 sensors in this seismic array are listed in Supplementary Table 2.

### Seismic detection methodology

The methodology we use for seismic event detection is based on the QuakeMigrate software package[13], designed for the automatic detection and location of earthquakes using waveform migration and stacking. Using continuous seismic data recorded on the Malyn seismic array, we transform the data at each sensor (and each channel for the case of the three-component sensor AKBB) into onset functions using the STA/LTA to help identify P-wave or S-wave seismic arrivals. For the detection of P-waves, we first apply a two-pole Butterworth bandpass filter with corner frequencies of 6 and 16 Hz, whereas for the detection of S-waves, we use a frequency band between 6 and 14 Hz. For both phases, we use a STA window of 0.3 s and a LTA window of 3 s. Using a lookup table of precomputed seismic travel times, we migrate and stack these onset functions over a grid of candidate locations at each time step. For an explosion that has been successfully recorded on the seismic network, the amplitudes of the onset functions will sum coherently (or coalesce) for the grid point corresponding to the source location and the time step matching the origin time of the explosion.

Our grid search is performed in two dimensions at the Earth's surface, between latitudes of 50° and 52° N and longitudes of 28° and 32.3° E, using a 1-km grid spacing. The theoretical P-wave and S-wave travel times for this grid are computed using an eikonal solver from NonLin-Loc[34]. As input to the eikonal solver, we use a 1D velocity model extracted from the 1 × 1 degree global crustal model, CRUST1.0 (ref. 35), at a longitude of 29.5° E and a latitude of 50.5° N (see Extended Data Table 1).

To generate event triggers, we apply a static detection threshold of 3.0 to the coalescence values generated at each time step in the migration and stacking. This threshold value was chosen to be deliberately low to ensure a high detection level. Although above the background noise level, this threshold also generates many false positives for signals originating outside our grid search, which become aliased to within our search area during the migration. To reduce the number of false positives, we remove all triggers that have locations originating in Belarus or that are mapped within the footprint of the array, in which many of the false-positive events become aliased.

We then rerun the migration and stacking on the remaining triggers using a modified STA/LTA function that better approximates a Gaussian for the onset functions, resulting in more precise event locations and origin times. Automatic magnitudes are also computed during this stage. A final manual screening of the events is performed by reviewing both the filtered waveforms with their theoretical arrival times and the migration images showing the coalescence values mapped onto our 2D regional grid. Any clear false positives are removed from the event catalogue during this manual screening. It should be noted that both the spatial filtering and manual screening are not required with a higher detection threshold, but the true detection rate would be reduced. We present a detection sensitivity analysis for three selected days in a subsequent section of Methods.

### Location uncertainty analysis

The location uncertainty of the migration algorithm can be assessed in terms of the migration images that are computed for each event. These 2D images show the coalescence function for a given time window, which is the basis for the location and origin time for each detection. It is first worth considering the theoretical location uncertainty for different locations in our target area. This can be achieved by computing point spread functions to show the impulse response for the migration. This theoretical response is influenced by the imaging function, the network geometry, the velocity model, the event location and the seismic phases we use to generate our images. We show point spread functions for two locations with very different distances to the array (more locations are provided in the Supplementary Information). Also, we show examples of observed explosions at similar locations to compare the synthetic case with that observed. In Extended Data Fig. 1, we show point spread functions and observed data from Malyn, approximately 9 km from the AKBB reference station. In Extended Data Fig. 2, we show examples from Chernihiv in the northeast, which is approximately 170 km from the AKBB reference station. To illustrate the necessity and influence of using both P-wave and S-wave onsets in the migration, we compute point spread functions for three cases: (1) when we have only the P-wave; (2) when we have only the S-wave; and (3) when we have both the P-wave and the S-wave. To quantify the location uncertainty within the point spread functions, we computed uncertainty ellipses based on the full width at half maximum. We perform a principal component analysis on all points that fall within 50% of the maximum value of the point spread function to generate an uncertainty ellipse, which is then centred on the image maximum. Such an approach has its limitations, as demonstrated by the uncertainty ellipses generated for the Chernihiv point spread functions (Extended Data Fig. 1a). For the P-wave only and S-wave only point spread functions, the values within 50% of the maximum are limited by the size of the migration area, which has the effect of artificially reducing the size of the uncertainty ellipse. Despite this limitation, we observe that the optimal imaging response (with both P-waves and S-waves across the network) provides a location precision of 4.7 km (semimajor axis) for an event at Chernihiv and 2.9 km for an event at Malyn, in which the distance to the array is greatly reduced. We also observe that, with only a single phase type (particularly only the P-wave), there is not sufficient resolution to locate events, especially at large distances or to the northwest and southeast of the array, at which azimuthal coverage is reduced. In the observed data from Malyn (Extended Data Fig. 1b), although there is now increased noise, and there are epistemic uncertainties included in the migration, the resulting migration image can still be regarded qualitatively as showing high precision owing to the clear P-wave and S-wave onsets in the waveform data. For Chernihiv, we show observed data (Extended Data Fig. 2b) that is of much poorer quality, with far fewer impulsive signals and higher noise in the waveform data, further degrading the imaging resolution that is inherent for that location. This poorer data quality is because of increased distance from the network, the single frequency band that is applied to all events in the imaging region and potential path effects. Although we observe a high azimuthal uncertainty, we are still able to constrain the distance relatively well owing to the P-wave and S-wave observations.

We show further observed examples from our event catalogue in Extended Data Fig. 3, to demonstrate the effect of location and data quality on the migration images. For example, in Extended Data Fig. 3a, we show an event typical of the high quality we observe in the region up to 50 km northeast of the array. This generates a high-resolution migration image. As a comparison, we also show a much lower quality event from a similar location in Extended Data Fig. 3b. For all examples, there is generally a good fit between the theoretical P-wave and S-wave arrival times and the observed arrivals in the waveforms.

### Detection sensitivity analysis

To demonstrate the detection sensitivity and to justify our choice of trigger threshold, we calculate the true positive rate (TPR) and false discovery rate (FDR) for three different days. The TPR is defined as,

$$\text{TPR} = \frac{\text{TP}}{\text{P}} = \frac{\text{TP}}{\text{TP} + \text{FN}}$$

in which TP is the number of true positives that have been detected and located by the migration algorithm and P is the total number of real positives in the dataset, that is, the total number of events that could theoretically be detected and located, which includes the total number of true positives and the total number of false negatives (FN), that is, events that the migration has been unable to reliably detect and locate. To verify marginal true positives (for which the signal-to-noise ratio is low) and to identify false-negative events (not detected by the migration algorithm), we manually screen all waveform data for each of the three days investigated. Although there are events in the dataset in which only single phases are observed across the network, this is not sufficient to provide a reliable event location, as demonstrated by the point spread functions in Extended Data Figs. 1 and 2. Such events are therefore regarded as false positives (FP), which are detected and mislocated by the migration algorithm. Similarly, in estimating the number of false negatives by manually screening the waveform data, we only consider events that can be manually picked and located using a traditional arrival-time-based location algorithm (HYPOSAT[36]) as potential false negatives. Events with extremely low signal-to-noise ratio or with only the P-wave present are not regarded as false negatives.

Because we wish to maximize the number of true positives while minimizing the number of false positives, we also calculate the FDR, which is defined as,

$$FDR = \frac{FP}{FP + TP}.$$

As the input to our migration algorithm are onset functions generated from the STA/LTA, assuming the trigger threshold is sufficiently above the noise floor, false positives arise from one of three scenarios. First, seismic events from outside the migration area can be spatially aliased into the region. Second, seismic events may have too few onsets across the network to reliably resolve the event location, but still be above the trigger threshold owing to high signal-to-noise ratios for a subset of phases or stations. Third, several events at different locations occurring simultaneously, or events that occur in short succession, resulting in onset functions for two different events being migrated as a single event. For example, the onset functions from the P-waves for event 1 may be migrated with the onset functions from the S-waves for event 2, generating an incorrect event.

The three days we selected for this analysis were 3 February, 7 March and 20 May 2022. 3 February was chosen as it represents a day before the invasion, for which only quarry blasts were detected and thus provides a good baseline measure. 7 March represents the day with the highest number of explosions detected and 20 May was selected because it is after the main Russian withdrawal from the region, yet there were still targeted attacks that were detected. Moreover, 20 May is a day with a high number of repeating signals that we commonly observe throughout our study period and which substantially contributes to the number of false positives detected. We believe these repeating signals to originate from possible mining activity in Belarus, yet are repeatedly aliased into our migration region.

In Extended Data Fig. 4, we show the time series of the maximum coalescence from across the migration region for each of the three days for which the triggering is performed. The repeating signals from the assumed mining activity from Belarus can be clearly observed between approximately 16:40 and 18:15 UTC and again from approximately 19:25 to 21:20 UTC on 20 May. The coalescence values corresponding to these events are similar to the values from the military attacks observed on this day.

In Extended Data Figs. 5–7, we show the TPR and the FDR as a function of triggering threshold. The total number of true positives and false positives are shown in the Supplementary Tables 3–5. For 3 February (Extended Data Fig. 5), for which we observe only three explosions related to quarry activity in the region, we observe a TPR of 100% between a trigger threshold of 2.4 and 4.0, whereas the FDR reduces from 99.4% to 40% over the same threshold range. Although the FDR generally decreases with a higher trigger threshold, we also observe some increases. For example, between thresholds of 2.9 to 3.0, there is an increase in FDR from 40% to 62.5%. This is caused by the duration of the coalescence values exceeding the threshold. At lower thresholds, the coalescence may exceed the threshold but not drop below it within a given time interval, meaning that there is no detrigger. At a higher threshold, within the same period, it is possible that the coalescence becomes triggered and detriggered several times, resulting in more false positives.

For 7 March (Extended Data Fig. 6), for which there are a total of 74 valid events, we observe an increase in the TPR from between 75.7% at a threshold of 2.4 to a maximum of 94.6% at thresholds of both 2.9 and 3.0. This increase in TPR despite a higher threshold is because of the influence of the false positives. For example, with several explosions, false positives can be generated owing to incorrect association of the different phases between the events, which will prevent the correct events being identified. Thus, with higher false positives, the number of true positives may also be reduced.

It is also worth noting that, although the number of true positives observed for 7 March never exceeds 70, we detected a total of 73 unique events across the different thresholds out of the possible 74 events. The single false-negative event that was only observed in the waveform screening was not detected by the migration owing to a lower frequency band being required for its detection.

For 20 May (Extended Data Fig. 7), the influence of the activity from Belarus is apparent in the FDR. Although the TPR reaches a maximum of 80% at thresholds of 2.7 and 2.8, it drops from 70% to 53.3% going from a threshold of 3.0 to 3.1. However, the FDR remains above 90% until a threshold of 3.5. This means that we are unable to minimize the number of false positives during this time period without severely affecting the TPR.

Although our aim is to maximize the number of true positives whilst minimizing the FDR, based on the TPR and FDR for the three selected days, there is no clear optimal threshold. The results from 3 February show that a threshold of 4.0 would provide the highest TPR (100%) while minimizing the FDR (40%). By contrast, for 7 March, when we observe the highest number of explosions, a lower threshold of 3.0 would be required to allow us to achieve the maximum TPR of 94.6%, resulting in an FDR of 58.1%. For 20 May, we would be required to further reduce the threshold to 2.8 to maximize the TPR (80%), but we then experience an FDR of 96.4% at this level.

Because many of the false positives that arise from the repeating Belarusian signals are generally aliased either into locations in the array or in Belarus, we choose to select a relatively low threshold of 3.0 as stated in the 'Seismic detection methodology' section and apply the aforementioned spatial filtering and final QC to remove the false positives.

### Infrasound detection methodology

The detection of infrasound phases is performed after the seismic detection and localization stage presented in these Methods. We assume that observed infrasound arrivals correspond to lower-tropospheric infrasonic propagation close to the surface, which is valid at close distances from the source (<100 km).

We search for infrasound arrivals for each event $e$ in time windows $w_e$ constrained by the event distance to each sensor $d_{e,s}$ (km) and origin time $t_{0,e}$, using realistic adiabatic sound velocities $c$ (km s$^{-1}$) between $c_{min} = 0.325$ km s$^{-1}$ and $c_{max} = 0.37$ km s$^{-1}$, such that $w_e = \{min_{s \in sensors}(d_{e,s}/c_{max}), max_{s \in sensors}(d_{e,s}/c_{min})\}$. To account for the candidate event time, we take the product of the waveform envelopes recorded at each sensor and a Gaussian function $g_{e,s} = e^{\{(t-\mu_{e,s})/\sigma\}^2}$, in which $t(s)$ is the time, with mean value $\mu_{e,s} = t_{0,e} + d_{e,s}/c(s)$ and standard deviation $\sigma = 7.5s$ for each candidate velocity $c$. We consider ten different candidate velocities

uniformly distributed in the range ($c_{min}$, $c_{max}$) to account for uncertainties in event location and surface sound velocities. Envelopes convolved with a Gaussian function are then linearly stacked to extract the stacking maximum $s_{max}$ used to identify infrasound arrivals. For each event, only the entry with candidate velocity $c$ showing the maximum value for $s_{max}$ is kept in the database as a potential detection.

To confirm the detection of infrasound arrivals, we apply a series of static detection thresholds, $t_{stack}$, on the stacking maximum, $s_{max}$, on the ratio of average maximum to average standard deviation across the array $t_{SNR}$ and on the ratio of maximum standard deviation to average standard deviations across the array $t_{std}$. Thresholds are chosen empirically after spot checking the detected waveforms such that $t_{stack} = 2.2 \times 10^{-8}$, $t_{SNR} = 3.655$ and $t_{std} = 7.1$. These values correspond to conservative choices to reduce the number of false positives.

### Automatic magnitude computation
We compute automatic local magnitudes using the QuakeMigrate software package[13] during the final migration and stacking stage described above. We remove the instrument response from each sensor and filter the data to simulate the response of a Wood–Anderson seismometer. To establish phase arrival times, we run an autopicker for each event by fitting a 1D Gaussian to the onset functions, in which the onset function exceeds the median absolute deviation outside the picking window by a factor of 8. We filter the instrument-corrected data with a 1–8-Hz bandpass filter and measure the maximum S-wave amplitude in a 4-s window from the automatic S-wave arrival time. Noise is estimated by measuring the root-mean-square amplitude in a 5-s window before the P-wave signal window. S-wave amplitudes that exceed the noise amplitude by a factor of 3 are then used to compute the local magnitude using the Hutton–Boore attenuation curves[37], with the mean value across all sensors used to compute a final network magnitude.

### Estimating explosive yield from seismic magnitudes
We estimate explosive yield from the automatic magnitude estimates using two empirically derived estimates. The upper yield estimates are calculated on the basis of the relationship,

$$M_L = 0.8834\log_{10}W - 1.4221$$

in which $W$ is the explosive yield in kg. This is derived from land-based explosions with known charge size listed in Supplementary Table 6 and compiled in ref. 30.

The lower yield estimates are based on the relationship derived for the Novaya Zemlya nuclear test site by[29],

$$m_b = 0.75\log_{10}Y + 4.25$$

in which $Y$ is the explosive yield in kilotonnes. It should be noted that this relationship has been derived for body-wave magnitudes ($m_b$) but we have applied it to the local magnitudes ($M_L$) calculated for the Ukrainian explosions.

Both the lower and upper estimates of explosive yield can be regarded as having high uncertainties and should be used only as a guide to the relative sizes of the explosions.

A histogram showing the distribution of yield estimates derived from seismic magnitudes is shown in Extended Data Fig. 8.

### Estimating explosive yield from acoustic phases
Empirical relationships exist between the explosive yield and acoustic maximum amplitudes, referred as the Blast Operational Overpressure Model (BOOM)[33], or dominant frequencies, referred to here as the Revelle model[38]. Frequency-based estimates are generally less sensitive to the atmospheric variability compared with amplitude estimates. However, empirical models based on frequency inputs have been constructed from far-field (>500 km from the source) historical data of mostly atmospheric nuclear explosions that are markedly different, in terms of energy release, from the conflict-related explosions investigated here. By contrast, models based on amplitude inputs have used close-range stations (<50 km from the source), which is more realistic for our event dataset.

Because the Revelle model relies on stratospheric returns at much larger distances from the source, we only estimate acoustic-based yields using the BOOM model. The BOOM empirical yield estimates[39] are constructed as:

$$Y_{BOOM} = [e^{(L-103.1-B/5.3)/20}/\{((S/1013)^{0.556}) \times ((1/110)^{0.444}) \times (25/R)^{1.333}\}]^{\frac{1}{0.444}}$$

in which $L = 20 \times \log_{10}(p/2e - 5)$ (dB), with $p$ the pressure perturbation amplitude and $B = \arctan(3 \times (dv/dz) \times (R/c_a))$, with $R$ (km) the source–receiver distance, $dv$ (m s$^{-1}$) the maximum difference in the sound speed and the surface sound speed, $dz$ (km) the altitude at which $dv$ is observed and $c_a$ (m s$^{-1}$) the sound velocity at the ground. $Y_{BOOM}$ requires the pressure amplitude as input, which is not directly available from our seismic recordings. To produce pressure amplitude estimates, we consider the relationship $v_z = H_{\rho w}P$, in which $H_{\rho w}$ is the air-to-ground transmission coefficient. $H_{\rho w}$ for acoustic waves travelling along the surface[40] is such that:

$$H_{\rho w} = \frac{c_a}{2(\lambda + \mu)}\frac{\lambda + 2\mu}{\mu},$$

in which $(\lambda, \mu)$ are the ground Lame parameters. Because high-frequency air-to-ground acoustic transmission can be sensitive to the poorly constrained uppermost ground layers, we consider two scenarios: (1) 'rock', which corresponds to the seismic velocities presented in Extended Data Table 1 and density $\rho = 2.85 \times 10^3$ kg m$^{-3}$, and (2) 'sediment', which corresponds to a scenario with much lower shear velocities at the ground such that $v_p$, $v_s$ and $\rho$ are 2 km s$^{-1}$, 0.55 km s$^{-1}$ and $1.93 \times 10^3$ kg m$^{-3}$, respectively. Furthermore, the BOOM model ($Y_{BOOM}$) uses sound velocity gradients as inputs through $dv$ and $dz$. Yet, these inputs only have a second-order impact on the energy transmission compared with seismic velocity parameters. Therefore, we use the following arbitrary values $dv = 1$ m s$^{-1}$ and $dz = 1$ km. Because we have signals recorded at several stations, we build estimates $\tilde{Y}$ as averages across all stations.

Yield estimates using the BOOM model are presented in Extended Data Fig. 9. We observe large discrepancies between the two types of seismic velocity model, with a much larger energy transmission, that is, lower yields, in the case of sediments. Only the estimates using the 'sediment' seismic model qualitatively match the distribution of yield estimates based on the magnitude computations in Extended Data Fig. 8. Strong biases exist in the empirical estimates provided by the BOOM model, which was developed using a different range of source yields and source–receiver distances. This highlights the need for ground-truth data for yield calibration in future studies.

### Event location using manually picked seismic arrivals
As well as the automatic event detections, we built a further database of located events for the purpose of validation (about 800 events). In contrast to the automatic catalogue, the P-wave and S-wave arrivals were manually picked by analysts on all AKASG stations, for which a clear onset was visible. For S-wave arrivals, picking on the single three-component station of the array tended to give more confident time picks. For a few events, acoustic arrivals were also picked manually.

We then located the events using the HYPOSAT location software[36]. This software implements an iterative optimization procedure inverting travel times for epicentre and source time. The source depth was set to zero because we anticipated signals generated on the Earth's surface. The same seismic velocity model as for the automatic stacking location was used. An uncertainty of 1 s on both P-wave and S-wave

arrivals was selected before inversion. Events with acoustic arrivals have been located by considering an extra acoustic phase in the inversion, travelling at constant velocity $c_a$. Because $c_a$ is not accurately known, we consider five potential velocity candidates uniformly distributed in the range $c_a = 0.33 \pm 0.01$ km s$^{-1}$. We then select the solution with the largest posterior likelihood. An uncertainty of 5 s on acoustic arrivals was selected before inversion to account for the difficulty in picking the onset when arrivals are dispersed and/or low in amplitude. Examples of two events located with manually picked arrivals, which include acoustic arrivals, are shown in Fig. 4a,e.

### Extracting event reports from Liveuamap

We extracted all the reported events between February and November 2022 from the automatic event catalogue available at https://liveuamap. com/. Daily event catalogues can be downloaded from the website as .geojson files, which can be processed using the built-in JSON Python library. Because the event catalogue contains entries unrelated to direct military activity, that is, potential explosions, we filtered the original catalogue using the following methodology: (1) we removed repeated entries, that is, entries with the same description that share the exact same location and time of the day but with dates varying across several days; (2) we then kept only entries that included the keywords $k_{save}$ shown below; (3) and, finally, we removed entries that included the keywords $k_{remove}$ shown below.

List of keywords of interest:

$k_{save}$ = 'exercise', 'ceasefire violations', 'military activity', 'artillery', 'damaged', 'wounded', 'launched', 'crash', 'strike', 'clashes', 'targeted', 'targeting', 'projectile', 'exploded', 'shot', 'fight', 'bombed', 'burn', 'blown up', 'siren', 'airdrop', 'destroy', 'killed', 'attacking', 'dropped', 'target', 'hit', 'struck', 'boom', 'smoke', 'explosion', 'blow up', 'firing', 'damage', 'hit', 'shelling', 'shelled', 'escalation', 'injured'.

$k_{remove}$ = '@Maxar satellite', 'Embassy', 'statement', 'German chancellor', 'DDoS', 'procession', 'warning', 'another video', 'will not succeed', 'calls civil', 'call civil', 'unexploded', 'repair', 'driving', 'treated', 'visit', 'take cover now!', 'says', 'telegram', 'found dead', 'satellite imagery', 'more footage', 'still no strikes', 'death toll', 'Ukrainian FM', 'frigate', 'seized', 'negotiation', 'phone call', 'minister', 'commander-in-chief', 'Zelensky', 'advisor', 'president of Ukraine', 'evacuation'.

### Data availability

The Malyn array is part of CTBTO's IMS. IMS data are available to State Parties through their National Data Centres. Access to all IMS data can also be granted on request using the virtual Data Exploitation Centre (vDEC) at https://www.ctbto.org/specials/vdec. The full catalogue of automatically detected explosions with origin time, location, local magnitude and yield estimates is available in Supplementary Table 1. We also indicate the detection of acoustic phases and the acoustic yield estimates in the catalogue. Waveform data for each event, the instrument response data and the contents of the supplementary information are all publicly available from the Open Science Framework (https://doi. org/10.17605/OSF.IO/PKAUV).

### Code availability

The QuakeMigrate software used to generate the automatic event catalogue is available from Zenodo (https://doi.org/10.5281/ zenodo.4442749). The HYPOSAT software that was used to manually relocate the two events shown in Fig. 4a,e is available from https://doi. org/10.2312/GFZ.NMSOP-2_Downloads.

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

**Acknowledgements** We thank the staff of the Main Centre of Special Monitoring in Ukraine for the continued operation of the Malyn seismic array throughout the war. This work has been supported by funding from the Norwegian authorities. At NORSAR, V. Oye provided invaluable support and gave feedback that led to notable improvements in this work. We would also like to thank colleagues at NORSAR who rapidly set up continuous data access that enabled the near-real-time monitoring of Ukraine. The views expressed herein are those of the authors and do not necessarily reflect the views of the CTBTO Preparatory Commission nor the views of the Norwegian government.

**Author contributions** B.D.E.D. designed and led the research, with input from all authors. B.D.E.D. adapted and implemented the stacking/migration method and processed the seismic dataset, with quality control support from B.P.G.-A. B.P.G.-A. and A.K. manually identified and hand-picked seismic arrivals that helped guide the initial parameterization of the stacking/ migration method. Q.B. performed the acoustic signal detection and analysis. Q.B. and B.D.E.D. extracted and processed reports from Liveuamap for validation. B.D.E.D., Q.B. and B.P.G.-A. drafted the initial manuscript. B.P.G.-A. created the figures in the main text and B.D.E.D. created the extended data figures. A.L. was responsible for data acquisition from the Malyn array. All authors took part in discussions and data analysis, contributed to the manuscript, contributed to Methods, performed a full interactive review of the original manuscript and approved the submitted version.

**Competing interests** The authors declare no competing interests.

**Additional information**
**Correspondence and requests for materials** should be addressed to Ben D. E. Dando.

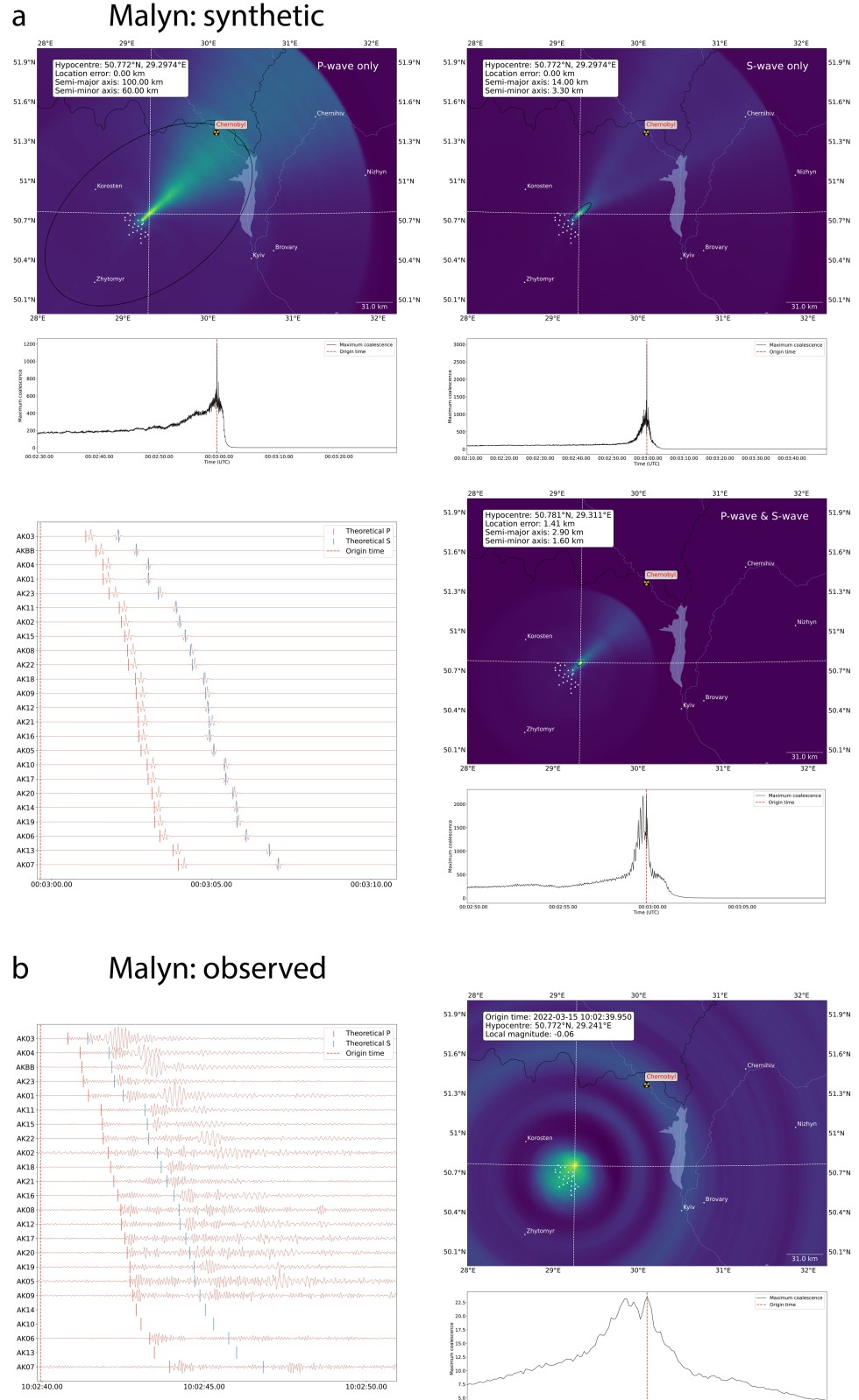

**Extended Data Fig. 1 | Synthetic and observed migration images for an event from Malyn. a**, The synthetic case, including the migration images (or point spread functions) for P-wave only (top left), S-wave only (top right) and P-wave and S-wave (bottom right), including the derived location (marked by the dashed lines) and an uncertainty ellipse based on the 50th percentile of the point spread functions. The synthetic waveforms used to generate the migration image are shown at the bottom left. **b**, An observed example from our catalogue showing the waveforms with the theoretical arrivals (left) and the migration image with the associated location (top right). For each case, we show the time series of the coalescence function used to derive the origin time.

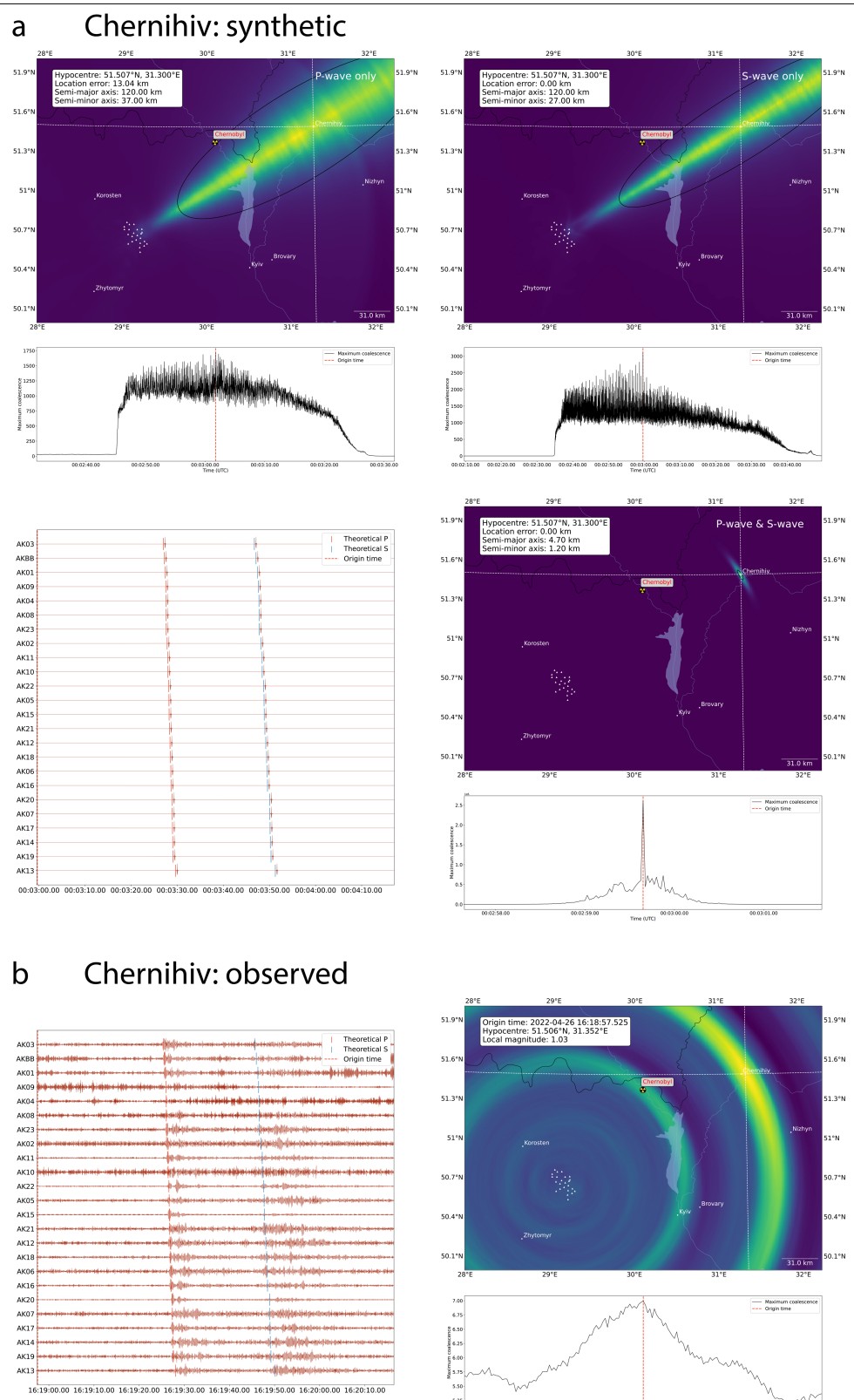

**Extended Data Fig. 2 | Synthetic and observed migration images for an event from Chernihiv. a**, The synthetic case, including the migration images (or point spread functions) for P-wave only (top left), S-wave only (top right) and P-wave and S-wave (bottom right), including the derived location (marked by the dashed lines) and an uncertainty ellipse based on the 50th percentile of the point spread functions. The synthetic waveforms used to generate the migration image are shown at the bottom left. **b**, An observed example from our catalogue showing the waveforms with the theoretical arrivals (left) and the migration image with the associated location (top right). For each case, we show the time series of the coalescence function used to derive the origin time.

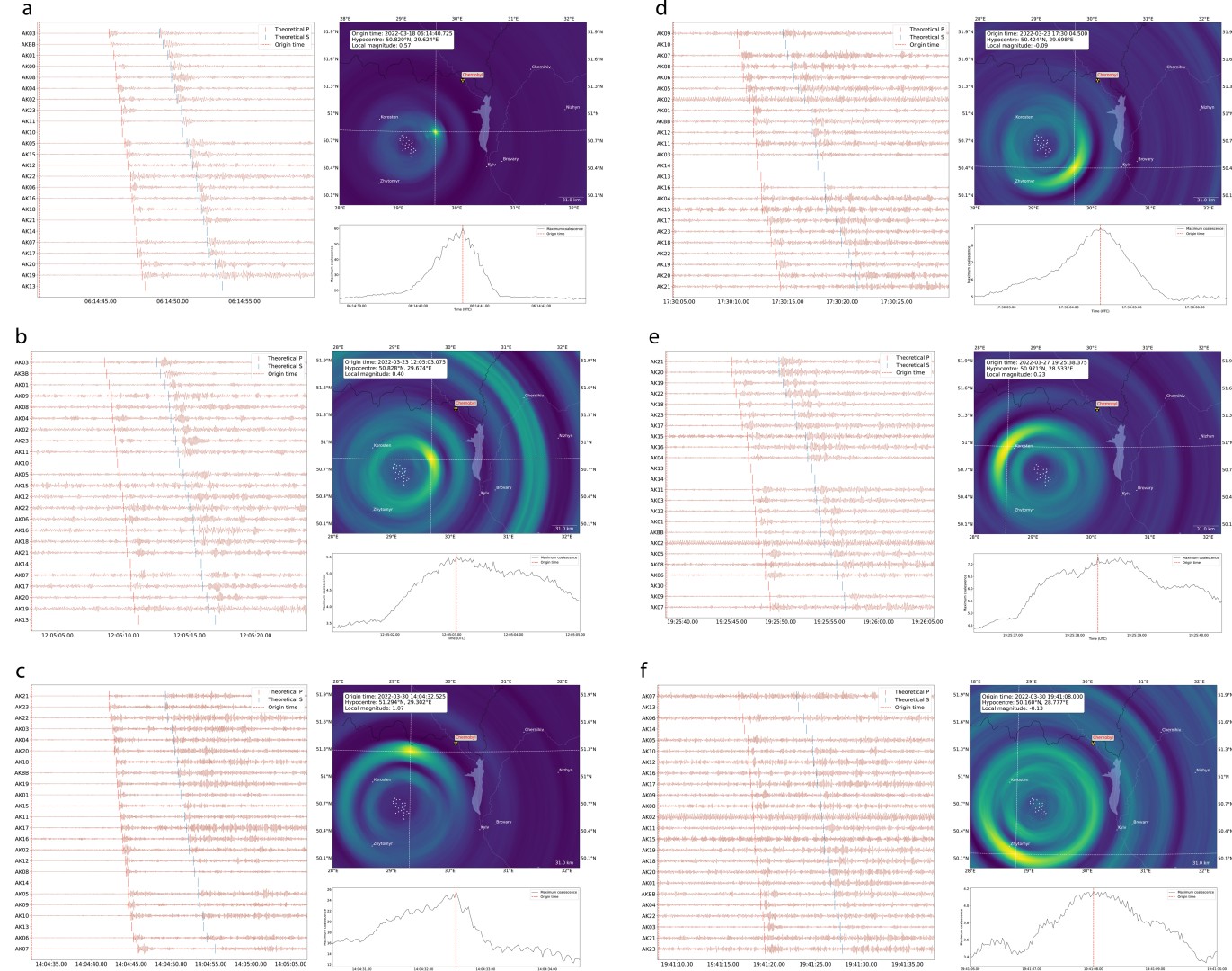

**Extended Data Fig. 3 | Examples of observed events and their associated migration images. a–f**, Each panel shows: the observed waveforms, including the theoretical arrival times of the P-wave and S-wave across the array (left), the migration image (coalescence function), including the derived location (top

right), and the time series for the coalescence function from which the event origin time is derived (bottom right). Panels **a** and **b** show two examples from a similar location with differing quality of data.

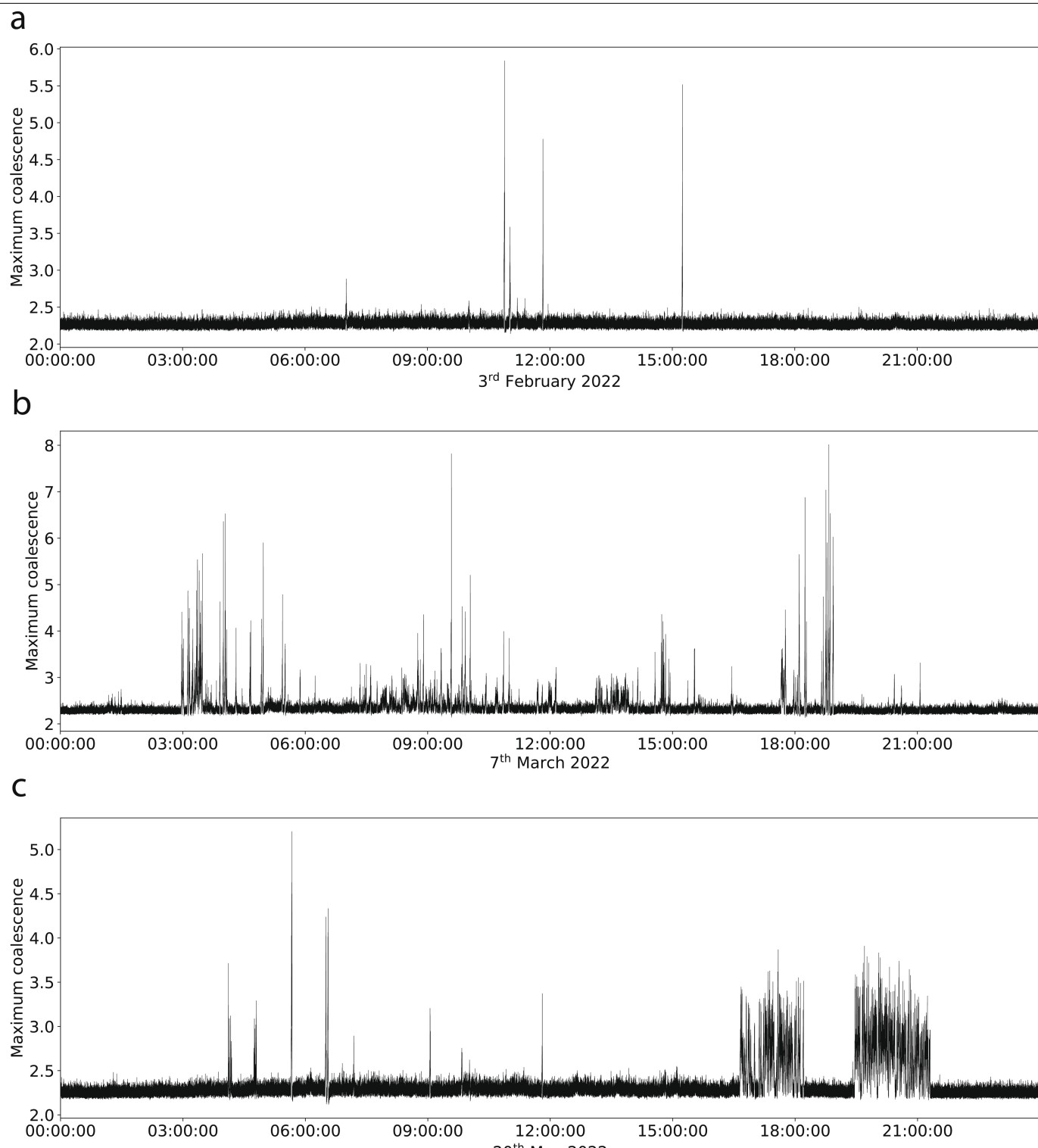

**Extended Data Fig. 4 | Maximum coalescence values for the three days in 2022 used in the detection sensitivity analysis.** Each subfigure shows the maximum coalescence value across the entire migration area, for each time sample throughout the day. **a**, 3 February 2022. **b**, 7 March 2022. **c**, 20 May 2022.

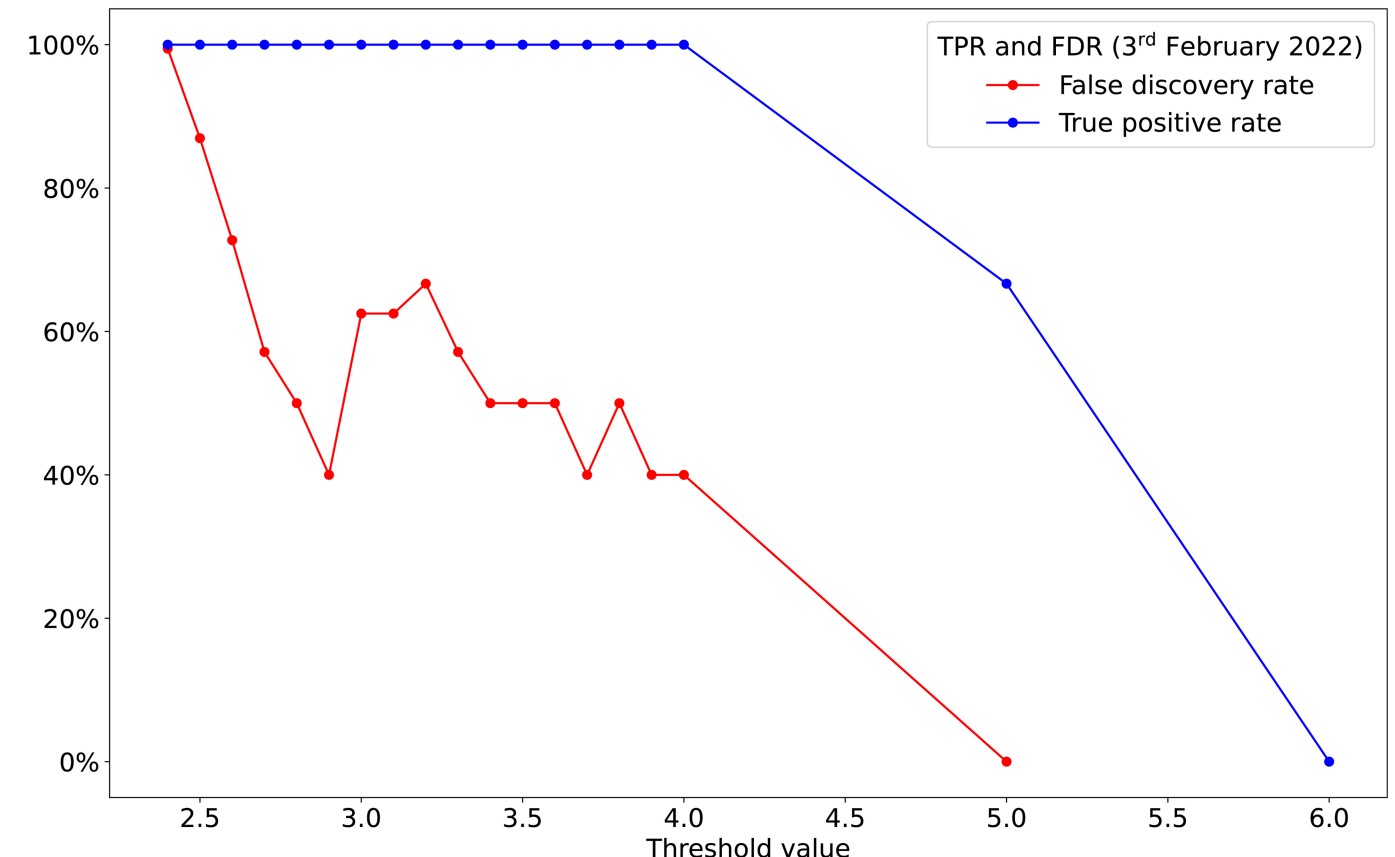

**Extended Data Fig. 5 | TPR and FDR for the migration algorithm on 3 February 2022.** A total of three true events were observed on this date. See Supplementary Information for the total number of true positives and false positives.

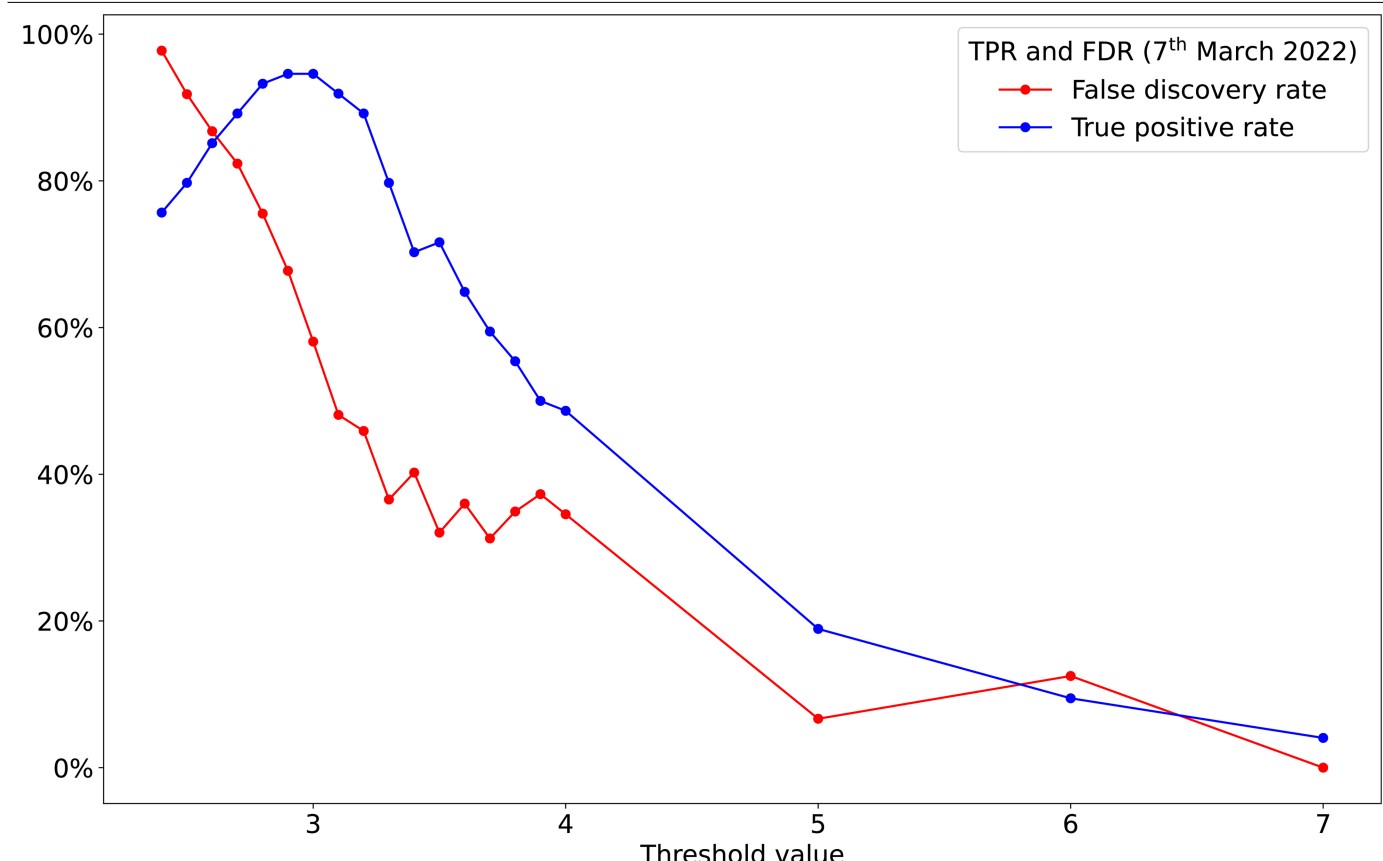

**Extended Data Fig. 6 | TPR and FDR for the migration algorithm on 7 March 2022.** A total of 74 true events were observed on this date. See Supplementary Information for the total number of true positives and false positives.

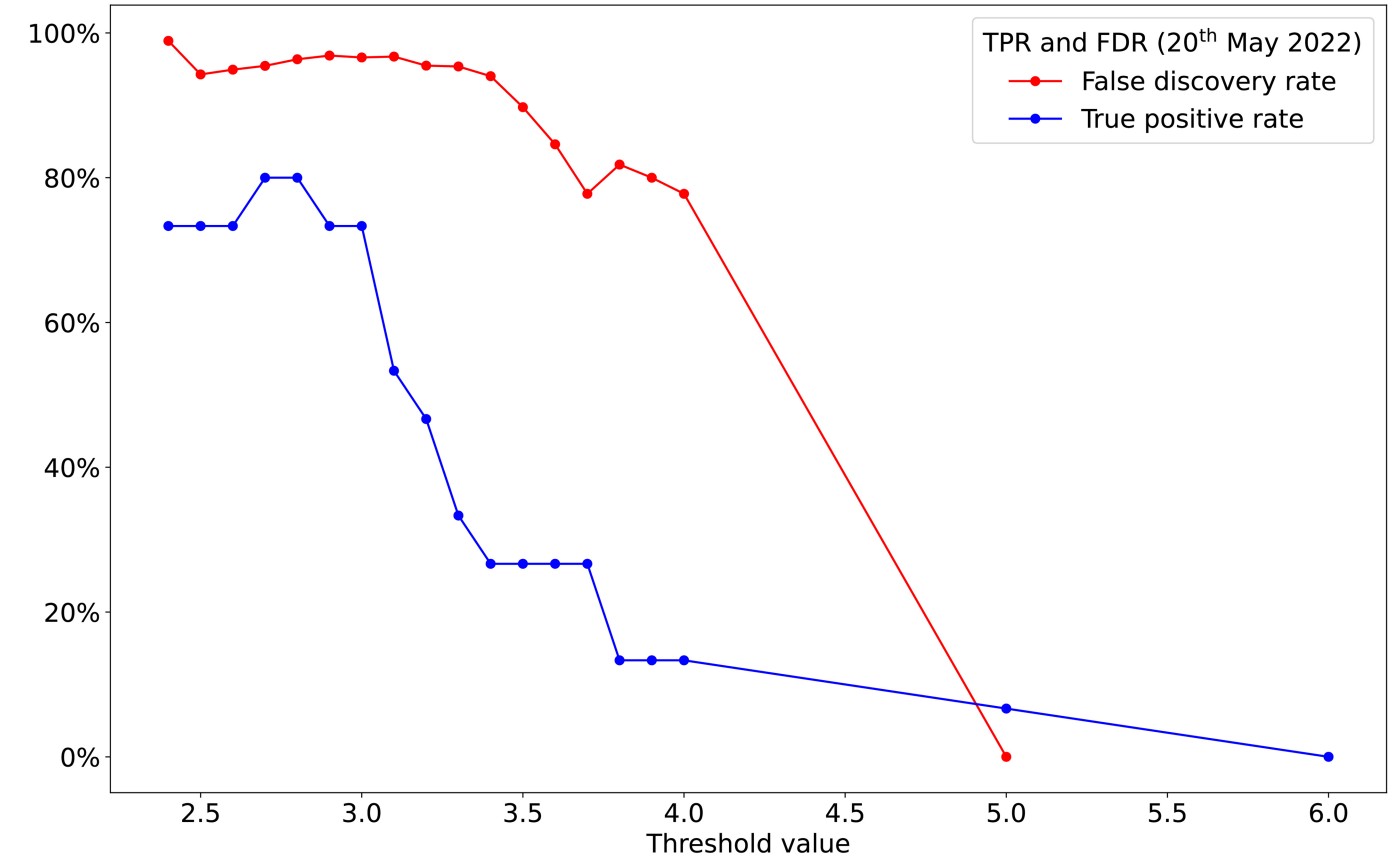

**Extended Data Fig. 7 | TPR and FDR for the migration algorithm on 20 May 2022.** A total of 15 true events were observed on this date. See Supplementary Information for the total number of true positives and false positives.

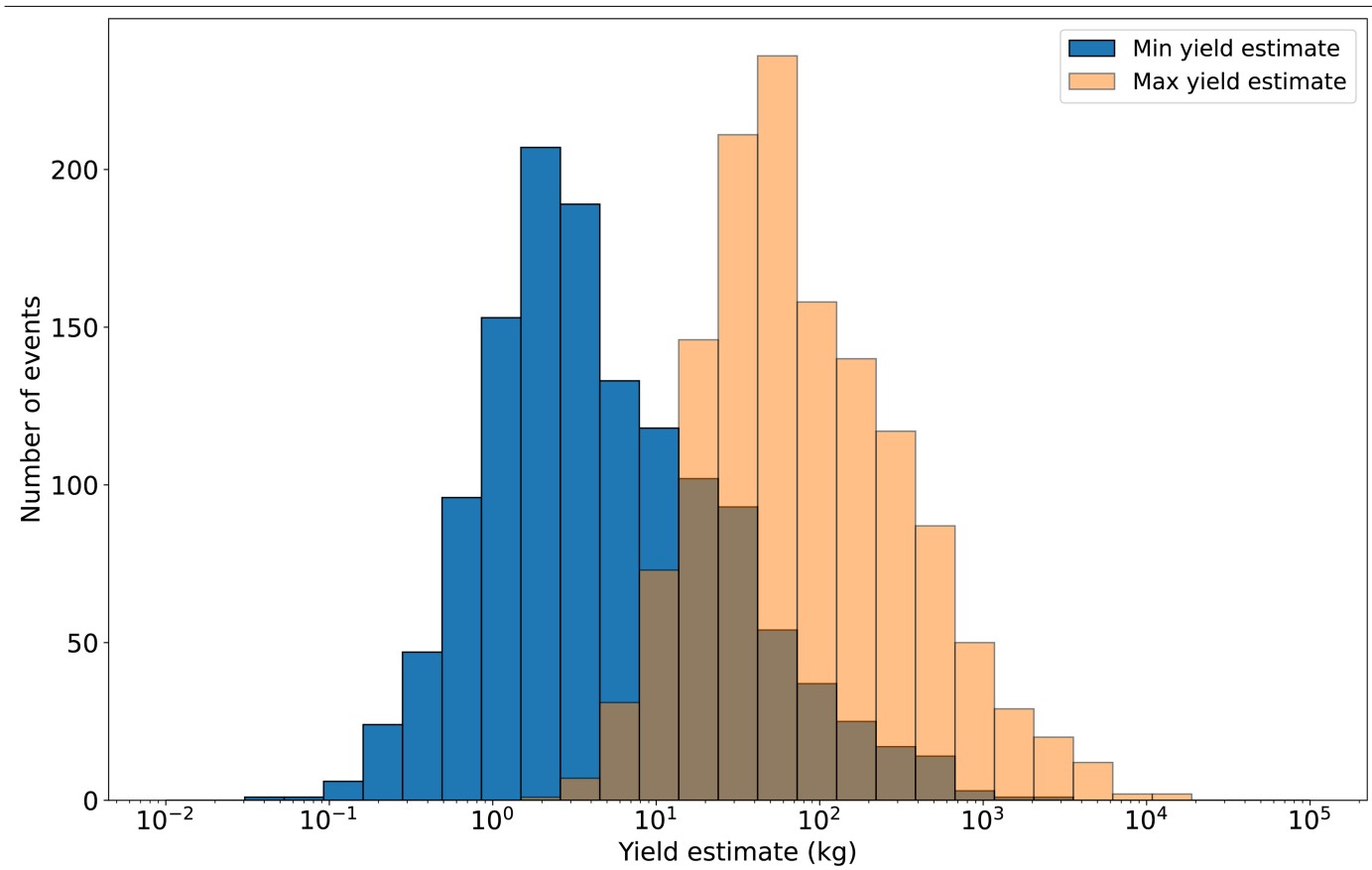

**Extended Data Fig. 8 | Histograms of the lower and upper yield estimates derived from the seismic magnitudes.** The upper yield estimates (peach) are based on a relationship derived from a catalogue of land-based explosions[30]. The lower yield estimates (blue) are based on the relationship derived for the Novaya Zemlya nuclear test site[22].

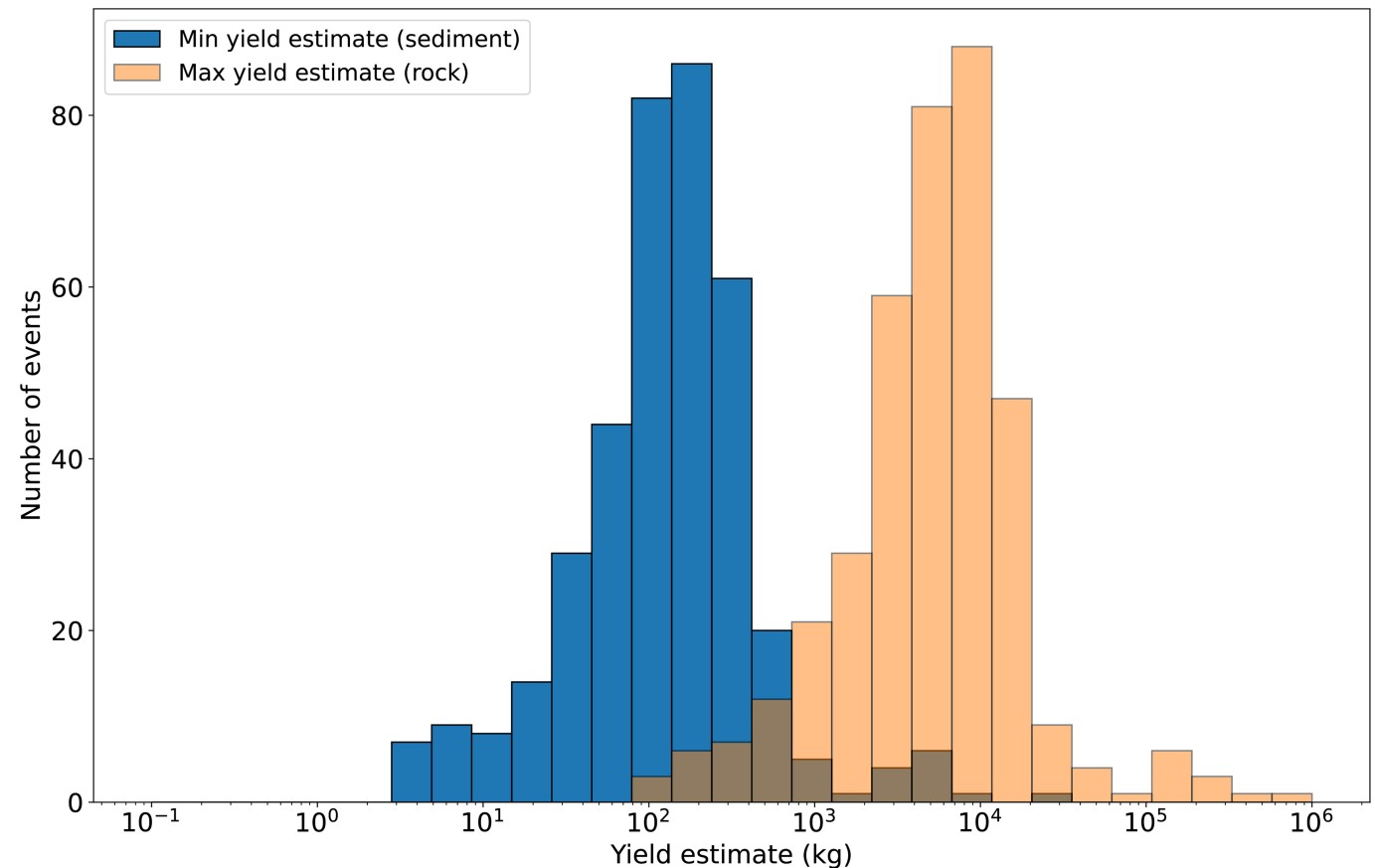

**Extended Data Fig. 9 | Histograms of the yield estimates derived from the BOOM model using two types of seismic velocity model.** The rock model (peach) uses the same seismic velocities as for event location (Extended Data Table 1), whereas the sediment model (blue) corresponds to much lower shear velocities (Methods).

**Extended Data Table 1 | 1D velocity model used to compute the travel-time table used for the automatic detection and location of explosions**

| Depth (km) | Vp (km/s) | Vs (km/s) |
|---|---|---|
| -0.17 | 6.20 | 3.60 |
| 13.39 | 6.40 | 3.70 |
| 26.94 | 6.80 | 3.90 |
| 40.90 | 8.32 | 4.61 |
| 100.90 | 8.32 | 4.61 |

**Extended Data Table 2 | Table of the manually refined locations shown in Fig. 4**

| Name | Origin time (UTC) | Latitude | Longitude |
|---|---|---|---|
| Hostomel airport | 2022-02-27 04:46:28.67 | 50.638 | 30.223 |
| Malyn train station | 2022-05-20 05:39:58.27 | 50.774 | 29.300 |