## [Peer Review File · Nature]

Manuscript Title: Identifying attacks in the Russia-Ukraine conflict using seismic array data

Reviewer Comments & Author Rebuttals

Reviewer Reports on the Initial Version:

Referee #1:

This paper is a first-of-a-kind investigation of conflict-related explosions using local seismic array data. Following a robust workflow, the authors are able to produce a catalogue of explosions occurring between February and November 2022 which shows a number of events exceeding the number of publicly reported attacks. The authors are experts in signal processing and event detection and, after reading the manuscript and methods part, I reckon that the data processing, event location and magnitude estimate is robust and state-of-the-art. I still have moderate comments and concerns though.

1/ I'm not an expert in seismic signal related to military operations but I'm wondering if a shot by itself (from a tank for example) can create an explosion significant enough to be detected and located? I guess it would be seen as a kind of precursor to the main explosion. Do you think shots could potentially be included in your catalogue as false positives? If you could detect the shots, that would prove useful for locating the position of the artillery. Could you please discuss that point?

2/ Also, do you think you can discriminate between missile strikes and explosions on structures like buildings or directly hitting the ground? I believe that direct impacts to the ground should show a more impulsive seismic signal? Do you have examples that could illustrate this and foster the interest for future studies on this topic?

3/ Finally, and most important, because you use a local array, your event detection capability decreases with distance between the array and the source. The most important value of a catalogue is its completeness meaning that, in a defined region, all events above a magnitude threshold should be detectable and included. I don't think this is the case in your catalogue. The catalogue for the eastern most region of Chernihiv is probably not as complete as for the region around Malyn, near the array, because of this detection bias. I think you should change the term "catalogue" to "list of detections" and further discuss how the magnitude of completeness reduces with distance from the array.

4/ The same applies to location uncertainties. Because you use a local array, location uncertainties increase with distance between the source and the array. Could you please add a discussion on that point?

Referee #2:

This paper uses seismic array data to detect and locate signals associated with the Russia-Ukraine conflict. The paper is of topical interest and is quite unique in the sense it applies seismic data to the problem of monitoring an active conflict. The only other similar study I am aware of explored seismic signals from Baghdad during the Iraq war (Aleqabi et al.) but was more limited in scope due to the fact it used a single seismometer. This study uses a large array, which allows the detection and location of seismic events (i.e., the construction of a seismic event catalog). This catalog is subsequently correlated with open-source information on military movements. I believe the paper would be of interest to a wide community, but it is unclear to me that this fits in Nature. The originality of the paper is the tie between a seismic catalog and an active military conflict. The catalog certainly does provide objective information about the conflict, but that information is

challenging to interpret without other sources of data. Adding to the challenge in interpretation, there is intrinsic uncertainty in seismic estimates of location and yield. I don't think the paper goes far enough in providing estimates of these uncertainties, or in providing sufficient estimate of what fraction of events themselves are legitimate. To improve these estimates, I recommend the following:

1) Detection: The paper does not provide a quantitative assessment of the number of false events. I'd really like to see a more formal analysis in the Supplementary information to quantify this (e.g., Receiver Operating Characteristic curves).

2) Location: It would be useful for the reader to understand the uncertainties in the event locations, which are only presented as points in Figures 1 and 3. Perhaps the reverse time migration volumes could be presented as Supplementary information for representative events occurring in different event clusters in Figure 1.

3) Yield estimation: The uncertainty here is systemic and hard to quantify. The upper and lower bounds used are not formal estimates of uncertainty. I am uncomfortable with statements like 'Yield estimates from the signals were also able to provide values consistent with a land attack cruise missile strike' for this reason. While the word 'consistent' implies uncertainty, I feel this statement could be misleading for non-experts. We are some way off being able to interpret seismic signals to specific military weapons without independent evidence (e.g., that a cruise missile was fired).

Regarding explosion size, recent work has developed models for seismic observations from surface explosions (e.g., Ford et al. doi.org/10.1785/0120130130, Kim and Pasyanos, doi.org/10.1785/0120220214, and the references therein). In my view, these provide a more direct tie between seismic observations and yield than via estimates of the seismic magnitude. It would be worth adding a few sentences referencing this body of literature, and why these models are not used).

Minor comments:

Line 17: Infrasound signals can also be detected by seismometers as air-to-ground coupled waves.

Line 60: Another relevant paper is Costley (2020), *Battlefield Acoustics in the First World War: Artillery Location*, *Acoustics Today*, 16, 31-39, [doi:10.1121/AT.2020.16.2.31](https://doi.org/10.1121/AT.2020.16.2.31)

Lines 161-162: The sentence 'Such an observation underlines the importance of seismic data for conflict monitoring' is problematic. If this statement is in reference to the relative 'importance' with respect to acoustic data, it is unfair (for the reasons given in the next sentence: we do not know the set of events that would be detected acoustically but not seismically). If it is a general statement of importance, it is incomplete. Perhaps the authors could rephrase to emphasize the relative benefits of seismic and acoustic data and the subsequent value on exploring both data types.

Line 377: The term 'likelihood' should be changed as this has a specific statistical meaning that is not consistent with the STA/LTA transform.

Line 509: $dz = 1$ (m/s) has a typo and should be dv .

Author Rebuttals to Initial Comments:

Referee #1:

This paper is a first-of-a-kind investigation of conflict-related explosions using local seismic array data. Following a robust workflow, the authors are able to produce a catalogue of explosions occurring between February and November 2022 which shows a number of events exceeding the number of publicly reported attacks. The authors are experts in signal processing and event detection and, after reading the manuscript and methods part, I reckon that the data processing, event location and magnitude estimate is robust and state-of-the-art. I still have moderate comments and concerns though.

1/ I'm not an expert in seismic signal related to military operations but I'm wondering if a shot by itself (from a tank for example) can create an explosion significant enough to be detected and located? I guess it would be seen as a kind of precursor to the main explosion. Do you think shots could potentially be included in your catalogue as false positives? If you could detect the shots, that would prove useful for locating the position of the artillery. Could you please discuss that point?

This is an important consideration which we agree needs clarifying. In the original text we stated,

"Explosions can correspond to either muzzle blasts, if the artillery is located near the array, ballistic shock waves, or ammunition explosions" with a citation of Dagallier et al (2019).

However, this gives the wrong impression that we are uncertain if our events originate from muzzle blasts (the shot), the ballistic shock wave, or the impact. It is possible to detect each of these components by acoustic sensing as demonstrated by the modelling performed in the Dagallier paper and in various acoustic/infrasound studies of military exercises. However, a muzzle blast itself will not generate significant seismic energy. For example, in Brissaud et al (2021) (<https://conferences.ctbto.org/event/7/contributions/802/>), live fire exercises by the Norwegian military were recorded by both seismic and infrasound sensors. The muzzle blast from a single shot could not be detected seismically beyond a distance of 11.6 km from the artillery. In a 2007 study from another field experiment (Anderson et al., 2007 - <https://doi.org/10.1117/12.738131>), different sized calibre shots were fired (60 mm to 120 mm) with seismic and acoustic sensors at 1.1 km distance from the artillery. It was only the largest calibre (120 mm) shots that could register the P-wave at this distance with an SNR of 5. In contrast, the infrasound signals from the muzzle blast can also be observed as air-to-ground converted waves on the seismic sensors, and these will typically propagate much larger distances. In the Brissaud et al. (2021) study, these converted waves from the muzzle blast are observed at up to 35 km from the artillery. Since these waves are travelling at acoustic velocities, the difference in moveout across the array means they would not generate false positives in the migration algorithm. As an aside, the converted acoustic waves that are generated from the impact detonation, which we observe for a subset of the events, propagate much larger distances due to the increased overpressure that they generate. These also do not generate false positives in the migration algorithm, again due to the large difference in moveout compared with the seismic arrivals.

In summary, since the muzzle blasts generate much less seismic energy than the impact detonation, and considering the large aperture of the Malin array, this means that it would be very unlikely that we would observe arrivals across the array that would generate detections above our triggering threshold. This is also backed up by the manual QC that was performed for our presented catalogue, which did not show any clear evidence of signals from muzzle blasts, such as stronger acoustic signals from the muzzle blast. We are therefore confident that our catalogue of events corresponds to impacts.

We have therefore removed the original text and added the following to the end of the 'Continuous monitoring of the Ukraine conflict using seismic data' section,

“While muzzle blasts, ballistic shock waves and the impact detonation can all generate infrasound energy (Dagallier et al, 2019), it is only the impact detonation that will likely generate sufficient seismic energy that can be observed at the distances we are monitoring (Anderson et al., 2007, Brissaud et al. 2021). The infrasound waves that we observe at the seismometers due to air-to-ground coupling, travel at much lower velocities than the seismic waves, which ensures they do not adversely affect our seismic detection algorithm. We are therefore confident that our detections most likely correspond to impact explosions.”

2/ Also, do you think you can discriminate between missile strikes and explosions on structures like buildings or directly hitting the ground? I believe that direct impacts to the ground should show a more impulsive seismic signal? Do you have examples that could illustrate this and foster the interest for future studies on this topic?

It would be fantastic to be able to make this distinction. We state in the manuscript that this dataset could be used in the future to help distinguish between different types of ammunition or artillery based on the signal characteristics, which in itself is challenging. Discriminating between types of impact or even fuse type (e.g. preimpact detonation fuses), would be another research avenue. However, we are not confident that we can make these distinctions now. We agree with the reviewer that the closer to the ground an explosion occurs, the more the high frequencies will contribute to your signal, i.e., the signal will be more impulsive. Additionally, the presence of the Rg phase and the ratio of body to surface waves may be indicative of ground explosions, since ground explosions are more likely to generate enough body wave energy propagating at critical incidence angles to constructively interfere as surface waves. However, source directivity will also likely play significant role in the excitation of waves. Finally, it's also notable that the research highlighted by referee #2 as well as being focussed on yield estimation, looks at constraining the height of explosions and this is something that needs to be further investigated in the future with these data.

3/ Finally, and most important, because you use a local array, your event detection capability decreases with distance between the array and the source. The most important value of a catalogue is its completeness meaning that, in a defined region, all events above a magnitude threshold should be detectable and included. I don't think this is the case in your catalogue. The catalogue for the eastern most region of Chernihiv is probably not as complete as for the region around Malyn, near the array, because of this detection bias. I think you should change the term “catalogue” to “list of detections” and further discuss how the magnitude of completeness reduces with distance from the array.

The catalogue around Chernihiv is certainly not as complete as the region around Malyn. In the original text we attempted to address this by writing,

“The most prominent activity is to the NE of Malyn, which while corresponding to a region where detection capability is high, also coincides with a region of intense fighting at the limits of the Russian-controlled territory during late February and March”.

However, we agree that this point should be further emphasised. We have added the following text to that same paragraph,

“Due to the detection bias close to the array, the magnitude of completeness reduces with distance from the array meaning the lowest magnitude explosions cannot be detected in locations such as Chernihiv, which is approximately 170 km from the array”.

With respect to changing ‘catalogue’ to being named a ‘list of detections’, we believe this would undermine the quality of the results and imply that it contains false detections rather than validated events. Since the results have all been quality controlled, we would prefer to refer to an event catalogue when discussing the presented results. A discussion of the quality control procedure is discussed within the ‘Seismic detection methodology’ in ‘Methods’, and a justification for this is presented in the newly added ‘Detection sensitivity analysis’ in ‘Methods’.

4/ The same applies to location uncertainties. Because you use a local array, location uncertainties increase with distance between the source and the array. Could you please add a discussion on that point?

This is an important consideration, and we fully agree that a discussion on the location uncertainties is needed. We have added an additional ‘Location uncertainty analysis’ section within the ‘Methods’. We have generated both point spread functions to show the theoretical imaging response at both Malyn and at Chernihiv, and compare the results to the observed data. We also demonstrate the necessity of including both P-wave and S-waves in the migration, without which there would be very limited distance constraint in the locations. It is for this reason that we typically do not keep detections where only observe single phases are observed. We also generate uncertainty ellipses for the point spread functions to help quantify the uncertainties. Although we only show two locations for the point spread functions, we repeated the calculation at a further 3 locations which we have attached for your assessment.

Lastly, we now show examples from a further 6 observed events at representative locations to demonstrate the variability in the data quality and the imaging response in our catalogue.

Referee #2:

This paper uses seismic array data to detect and locate signals associated with the Russia-Ukraine conflict. The paper is of topical interest and is quite unique in the sense it applies seismic data to the problem of monitoring an active conflict. The only other similar study I am aware of explored seismic signals from Baghdad during the Iraq war (Aleqabi et al.) but was more limited in scope due to the fact it used a single seismometer. This study uses a large array, which allows the detection and location of seismic events (i.e., the construction of a seismic event catalog). This catalog is subsequently correlated with open-source information on military movements. I believe the paper would be of interest to a wide community, but it is unclear to me that this fits in Nature. The originality of the paper is the tie between a seismic catalog and an active military conflict. The catalog certainly does provide objective information about the conflict, but that information is challenging to interpret without other sources of data. Adding to the challenge in interpretation, there is intrinsic uncertainty in seismic estimates of location and yield. I don't think the paper goes far enough in providing estimates of these uncertainties, or in providing sufficient estimate of what fraction of events themselves are legitimate. To improve these estimates, I recommend the following:

1) Detection: The paper does not provide a quantitative assessment of the number of false events. I'd really like to see a more formal analysis in the Supplementary information to quantify this (e.g., Receiver Operating Characteristic curves).

We thank the referee for identifying this and agree that a quantitative assessment is needed to understand the detection sensitivity of the method we have implemented. To address this, we have now included a detection sensitivity analysis in the Methods section, where we show the effect of different triggering thresholds for three separate days with different event characteristics. This analysis includes thorough manual screening for each of the 24-hour periods and quality control of the migration results at the different thresholds to ensure we correctly identify all possible events including true positives and false negatives. As part of the analysis, we show the true positive rates, and the false discovery rates prior to any of the post-processing quality control that we apply to the event catalogue.

We believe this analysis provides a good justification for our choice of threshold but also highlights the need for the spatial filtering and manual QC that we applied to the complete catalogue. We would like to emphasise that based on this approach we aimed to remove all false positives from the event catalogue presented in the manuscript.

2) Location: It would be useful for the reader to understand the uncertainties in the event locations, which are only presented as points in Figures 1 and 3. Perhaps the reverse time migration volumes could be presented as Supplementary information for representative events occurring in different event clusters in Figure 1.

We also fully agree that uncertainties in the event locations should be discussed and presented. We have approached this by adding an additional 'Location uncertainty analysis' section within the 'Methods'. Here, we first show point spread functions for two locations – Malyn, which is approximately 9 km from the array, and Chernihiv which is approximately 170 km from the array. For both of these synthetic cases we also compare the results to observed data from these same locations. The point spread functions show the theoretical imaging response for cases where only the P-wave or S-wave is observed as well as both phases. We do this to show the importance of requiring both P and S-wave observations in our detections. We also generate uncertainty ellipses for the point spread functions to help quantify the uncertainties. In addition, we show further 6 observed examples from our event catalogue for representative locations. This demonstrates the variability in waveform quality and imaging response. While we only include point spread functions for two locations in the manuscript, we have calculated point spread functions for a total of 5 locations which we have attached for your review.

3) Yield estimation: The uncertainty here is systemic and hard to quantify. The upper and lower bounds used are not formal estimates of uncertainty. I am uncomfortable with statements like 'Yield estimates from the signals were also able to provide values consistent with a land attack cruise missile strike' for this reason. While the word 'consistent' implies uncertainty, I feel this statement could be misleading for non-experts. We are some way off being able to interpret seismic signals to specific military weapons without independent evidence (e.g., that a cruise missile was fired).

Regarding explosion size, recent work has developed models for seismic observations from surface explosions (e.g., Ford et al. doi.org/10.1785/0120130130, Kim and Pasyanos, doi.org/10.1785/0120220214, and the references therein). In my view, these provide a more direct tie between seismic observations and yield than via estimates of the seismic magnitude. It would be worth adding a few sentences referencing this body of literature, and why these models are not used).

We agree that the statement on the yield estimate for the Malyn train station attack does not stand-up to scrutiny and we have removed it from the text.

We also thank the referee for the references linking seismic observations to yield estimation. As shown in these publications, significant work has been performed to develop reliable models for

yield estimation, which far exceed the simple empirical approach we have taken in this study. However, the justification for our approach is due to both technical limitations of the data, and the practicalities of implementing a real-time system, which is one of the aims of this paper. Firstly, from a data perspective, the Malin array comprises vertical component sensors on all but a single site (AKBB). The methods described in Ford et al (2014), Koper et al (2002), and Kim and Pasyanos (2023), all involve the processing of 3-component seismic data, which we do not have available beyond the single AKBB station. By itself, this station would not produce robust results. Furthermore, the waveform envelope method of Pasyanos et al. (2012), which would provide much more reliable yield estimates if horizontal-component data were available, requires careful calibration, including reliable code decay parameters that are region dependent, and that are not currently established. Lastly, these methods are arguably less well suited to automation in a real-time system, which is required to provide a rapid estimation of the size of the explosion.

Although we make clear in the text that our approach for estimating the yield has significant limitations, we agree that we should highlight the alternative methods. We have added the following text to address this,

“Estimating the explosive yield from seismic data is a challenging research area, with numerous approaches based on both empirical observations and physics-based models (Koper et al. 2002, Pasyanos et al. (2012), Ford et al. (2014)). Recent methods combining both seismic and acoustic observations (Kim and Pasyanos (2023)) show significant promise in resolving both yield and height of explosions. Since the Malin array comprises vertical component data on all but a single site, we are limited in the approach we can take. We focus on providing a rapid evaluation of the explosive strength by automatically computing seismic magnitudes (Methods).”

Minor comments:

Line 17: Infrasound signals can also be detected by seismometers as air-to-ground coupled waves.

We have changed the text to be,

“The large overpressure generated by an explosion, shakes the Earth’s atmosphere and subsurface, and the resulting ground motion can be recorded by seismometers. The infrasound signals that propagate through the atmosphere can be detected by microbarometers, or at seismometers as air-to-ground coupled waves”

Line 60: Another relevant paper is Costley (2020), Battlefield Acoustics in the First World War: Artillery Location, Acoustics Today, 16, 31-39, doi:10.1121/AT.2020.16.2.31

We have now added this reference.

Lines 161-162: The sentence ‘Such an observation underlines the importance of seismic data for conflict monitoring’ is problematic. If this statement is in reference to the relative ‘importance’ with respect to acoustic data, it is unfair (for the reasons given in the next sentence: we do not know the set of events that would be detected acoustically but not seismically). If it is a general statement of importance, it is incomplete. Perhaps the authors could rephrase to emphasize the relative benefits of seismic and acoustic data and the subsequent value on exploring both data types.

While we do not have much room in the text to expand on the relative benefits seismic and acoustic data, we have rephrased the last part of this paragraph to be more balanced. We removed the sentence starting ‘Such an observation...’ and now write,

“It is also worth noting that not all explosions observed in the waveform data feature seismic arrivals, with some events only detectable from their acoustic signature. Such events most likely correspond to explosions at higher altitude, at larger distances from the source (> 100 km), or with lower yield. These observations highlight that both acoustic and seismic monitoring can play an important role in conflict monitoring.”

Line 377: The term ‘likelihood’ should be changed as this has a specific statistical meaning that is not consistent with the STA/LTA transform.

We have changed,

“Using continuous seismic data recorded on the Malin seismic array, we transform the data at each sensor (and each channel for the case of the 3-component sensor AKBB) into onset functions showing the likelihood of a P-wave or S-wave seismic arrival, using the short term average to long term average amplitude ratio (STA/LTA).”

to

“Using continuous seismic data recorded on the Malin seismic array, we transform the data at each sensor (and each channel for the case of the 3-component sensor AKBB) into onset functions using the short term average to long term average amplitude ratio (STA/LTA), to help identify P-wave or S-wave seismic arrivals.”

Line 509: $dz = 1$ (m/s) has a typo and should be dv .

This has now been corrected.

Reviewer Reports on the First Revision:

Referee #1:

I acknowledge the authors for their careful and thorough revision. All my concerns and questions have been properly raised and I believe that the manuscript is now ready for publication.

Referee #2:

I am very satisfied with the careful responses to my concerns on the first submission. I believe the paper is in good shape to be published.